# A conserved RNA degradation complex required for spreading and epigenetic inheritance of heterochromatin

Gergana Shipkovenska[1], Alexander Durango[1], Marian Kalocsay[2], Steven P Gygi[2], Danesh Moazed[1]*

[1]Howard Hughes Medical Institute, Department of Cell Biology, Harvard Medical School, Boston, United States; [2]Department of Cell Biology, Harvard Medical School, Boston, United States

**Abstract** Heterochromatic domains containing histone H3 lysine 9 methylation (H3K9me) can be epigenetically inherited independently of underlying DNA sequence. To gain insight into the mechanisms that mediate epigenetic inheritance, we used a *Schizosaccharomyces pombe* inducible heterochromatin formation system to perform a genetic screen for mutations that abolish heterochromatin inheritance without affecting its establishment. We identified mutations in several pathways, including the conserved and essential Rix1-associated complex (henceforth the rixosome), which contains RNA endonuclease and polynucleotide kinase activities with known roles in ribosomal RNA processing. We show that the rixosome is required for spreading and epigenetic inheritance of heterochromatin in fission yeast. Viable rixosome mutations that disrupt its association with Swi6/HP1 fail to localize to heterochromatin, lead to accumulation of heterochromatic RNAs, and block spreading of H3K9me and silencing into actively transcribed regions. These findings reveal a new pathway for degradation of heterochromatic RNAs with essential roles in heterochromatin spreading and inheritance.

*For correspondence:
danesh@hms.harvard.edu

**Competing interests:** The authors declare that no competing interests exist.

## Introduction

Heterochromatic domains of DNA are a conserved feature of eukaryotic chromosomes and play central roles in regulation of transcription, inactivation of transposons, and maintenance of genome integrity (*Allshire and Madhani, 2018*; *Saksouk et al., 2015*). Heterochromatin is established by the recruitment of histone-modifying enzymes to nucleation sites, followed by spreading of the modification by a read-write mechanism, which involves recognition of the initially deposited modification and deposition of the same modification on histones in adjacent nucleosomes. The so-formed broad domains of repression can be epigenetically inherited through many cell divisions. Recent studies using inducible heterochromatin formation strategies have demonstrated that DNA sequence-independent and DNA sequence-dependent pathways work together with the histone modification directed read-write to mediate epigenetic inheritance of heterochromatin (*Audergon et al., 2015*; *Wang and Moazed, 2017*; *Yu et al., 2014*; *Laprell et al., 2017*; *Coleman and Struhl, 2017*; *Ragunathan et al., 2015*). However, the mechanisms that mediate faithful epigenetic inheritance are not fully understood.

In the fission yeast *Schizosaccharomyces pombe* heterochromatin is assembled at pericentromeric DNA regions, telomeres, the mating type locus, and the ribosomal DNA repeats (*Holoch and Moazed, 2015a*; *Wang et al., 2016*). These regions are associated with histone H3K9 methylation, a conserved marker of heterochromatin, which is catalyzed by the Clr4 (Suv39h) methyltransferase (*Rea et al., 2000*; *Nakayama et al., 2001*; *Bannister et al., 2001*). Studies using an inducible heterochromatin formation system demonstrated that in some cells heterochromatin can be inherited

independently of any input from underlying DNA sequence (*Ragunathan et al., 2015*; *Audergon et al., 2015*). In these studies, H3K9 methylation is induced by fusion of a Clr4 fragment, containing the methyltransferase domain but lacking the chromodomain (CD) required for recognition of H3K9me, to the bacterial Tetracycline Repressor (TetR-Clr4ΔCD), which binds to *tetO* arrays inserted at a euchromatic locus. The inheritance of the inducible heterochromatic domain depends on the read-write capability of Clr4 and can be readily observed in cells in which the anti-silencing factor *epe1⁺*, encoding a Jmjc family demethylase member, has been deleted (*Trewick et al., 2007*; *Bao et al., 2019*; *Sorida et al., 2019*; *Ragunathan et al., 2015*; *Audergon et al., 2015*). Mutation or deletion of the chromodomain of Clr4, required for efficient read-write, specifically disrupt epigenetic inheritance without any effect on heterochromatin establishment (*Ragunathan et al., 2015*; *Audergon et al., 2015*). In wild-type *epe1⁺* cells, in addition to the read-write mechanism, epigenetic inheritance of heterochromatin requires input from specific DNA sequences or a locally generated siRNA amplification loop (*Wang and Moazed, 2017*; *Yu et al., 2018*). The development of inducible heterochromatin domains, and the demonstration that they could be epigenetically inherited, provides a unique opportunity to delineate the pathways that are specifically required for heterochromatin inheritance.

To investigate whether other pathways work together with the read-write mechanism to maintain heterochromatin, we used the inducible heterochromatin system to conduct a genetic screen for mutations that abolish heterochromatin inheritance without affecting its establishment. Our screen identified mutations in several pathways, including mutations in known heterochromatin associated factors, components of the DNA replication machinery, and in three subunits of an RNA processing complex, composed of Rix1, Crb3, Grc3, Mdn1, Las1 and Ipi1 (*Castle et al., 2012*; *Schillewaert et al., 2012*; *Castle et al., 2013*; *Gasse et al., 2015*; *Fromm et al., 2017*). This complex, which we propose to name the 'rixosome', is conserved from yeast to human, has well-characterized essential functions in ribosomal RNA (rRNA) processing and has been previously shown to localize to heterochromatin in a Swi6-dependent manner (*Kitano et al., 2011*; *Iglesias et al., 2020*), but its role in heterochromatin function remains unknown. We show that the rixosome targets heterochromatic RNAs for degradation through the conserved 5′−3′ exoribonuclease pathway to clear a path for Clr4-mediated read-write.

## Results

### Epigenetic maintenance defective (*emd*) mutants

Since the requirements for establishment and maintenance of heterochromatin cannot be readily separated at endogenous sites, the identification of pathways specifically required for epigenetic inheritance of heterochromatin has not been possible. To identify such pathways, we employed an inducible heterochromatin system which allows for separation of the establishment and maintenance phases of heterochromatin formation (*Ragunathan et al., 2015*; *Audergon et al., 2015*; *Figure 1A*). Heterochromatin is established by the recruitment of a TetR-Clr4ΔCD fusion to a *10xtetO-ade6⁺* reporter in the absence of tetracycline. The TetR-Clr4ΔCD fusion is then released by the addition of tetracycline, which disrupts the interaction of the TetR domain with the *10xtetO* targeting sequence. The maintenance of silencing is monitored by the color of the colonies grown on low adenine medium, supplemented with tetracycline. *ade6^OFF* cells give rise to red colonies, due to the accumulation of a red intermediate in the adenine biosynthetic pathway, while *ade6^ON* cells give rise to white colonies. In the absence of the anti-silencing factor Epe1 (*epe1Δ*), growth on tetracycline-containing medium results in a variegated phenotype, due to the stable propagation of the heterochromatic domain in roughly 60% of cells (*Audergon et al., 2015*; *Ragunathan et al., 2015*).

We performed random mutagenesis by ethyl methanesulfonate (EMS) in cells carrying this reporter and screened for *epigenetic maintenance defective* (*emd*) mutants (*Figure 1B*). We plated the mutagenized cultures on low adenine medium lacking tetracycline and then replica plated the mutant colonies on low adenine medium containing tetracycline (*Figure 1—figure supplement 1A*). We screened the resulting replicas for colonies which lost the silencing of the *10xtetO-ade6⁺* reporter in the presence of tetracycline, but not in its absence. Such colonies represent mutants which are specifically defective for the maintenance of silencing. We screened 80,000 colonies and obtained 83 apparent *emd* mutants (*Figure 1—figure supplement 1B*). About half of these mutants

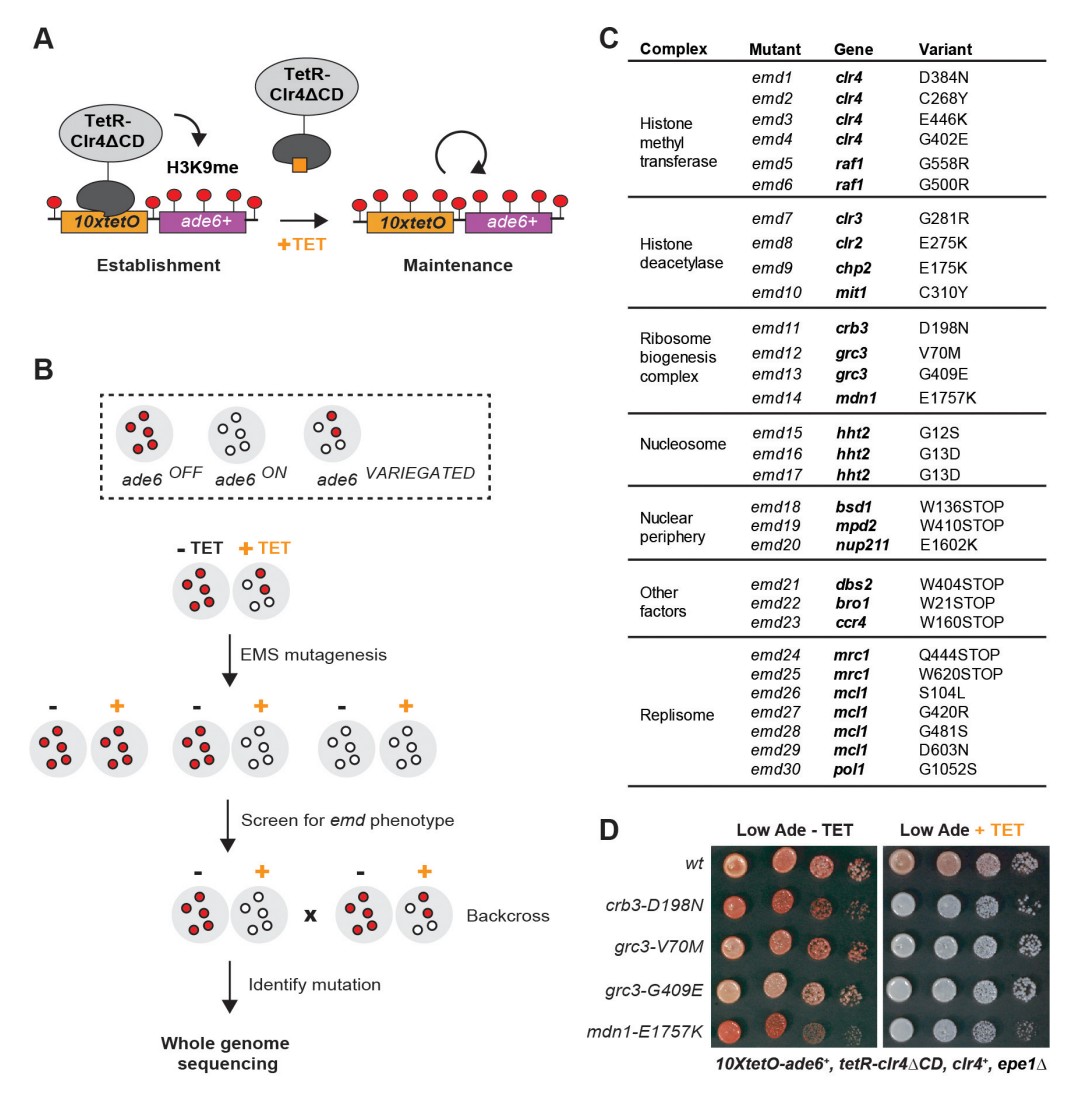

**Figure 1.** A Genetic Screen Identifies Factors Essential for Heterochromatin Inheritance. (**A**) Schematic diagram representing an inducible reporter system for heterochromatin establishment and inheritance (**Ragunathan et al., 2015**, **Audergon et al., 2015**). *10xtetO*, 10 copies of the *tet* operator; TetR-Clr4ΔCD, fusion of TetR and Clr4 lacking the chromodomain; H3K9me, di- and tri-methyl histone H3 Lys 9; TET, tetracycline. (**B**) Schematic diagram of the screen design. EMS, ethyl methanesulfonate; *emd*, epigenetic maintenance defective. (**C**) List of mutations identified by whole genome sequencing, with their respective genes and the protein complexes that they are known to associate with, where applicable. (**D**) Silencing assay on low adenine medium in the absence (-TET) or presence (+TET) of tetracycline for mutants *emd11-14*, carrying point mutations in subunits of the rixosome. Ade, adenine.

The online version of this article includes the following figure supplement(s) for figure 1:

**Figure supplement 1.** Screen for Isolation of Inheritance Defective Mutants.

showed single-locus segregation patterns in random spore analysis when backcrossed to the original reporter strain. The remaining mutants showed complex segregation patterns, consistent with cumulative weak effects from multiple contributing loci. We proceeded to characterize the *emd* mutants with single gene segregation patterns, as these likely represent the factors with the strongest effects on heterochromatin maintenance.

While all *emd* mutants lost silencing on tetracycline-containing medium, they differed in their ability to establish efficient silencing on medium lacking tetracycline (*Figure 1—figure supplement 1C*). In some mutants, the silencing of the *10xtetO-ade6⁺* reporter was slightly weaker than that of the non-mutagenized strain, indicating that the mutated genes likely play a role in heterochromatin

establishment, in addition to their role in maintenance. In the remaining mutants, the establishment of silencing was either as strong as or stronger than that of the non-mutagenized reporter strain, suggesting that the underlying mutations are specifically required for heterochromatin maintenance. The stronger than wild-type establishment may reflect weakened heterochromatin structure at the endogenous domains and the resulting redistribution of silencing factors from the endogenous to the ectopic domain.

To isolate the variants linked to the maintenance defect in each of the *emd* mutants, we performed whole genome sequencing combined with pooled linkage analysis (*Birkeland et al., 2010*; *Iida et al., 2014*). Out of the 40 *emd* mutants, 30 yielded high confidence single nucleotide variants in coding regions of the genome (variants present in >90% of the mutant spore pool), one had no coding variant despite good genome coverage, and the remaining 9 had insufficient sequencing coverage to be analyzed. The identified mutations mapped to 19 genes, most of which encode subunits of one of five complexes (*Figure 1C*). Among these are the CLRC and SHREC complexes, which play key roles in heterochromatin formation, and novel mutations in genes with no known function in heterochromatin maintenance. These included three independently identified mutations in the N-terminal tail of histone H3, encoded by *hht2,* and multiple mutations in the Mrc1, Mcl1 and Pol1 components of the DNA replication machinery (*Figure 1C*). In addition, we isolated mutations in multiple genes, encoding factors broadly associated with the nuclear periphery (*bsd1*, *nup211*, *mpd2*), a subunit of the major Ccr4-Not deadenylase complex (*ccr4*), a universal stress family protein (*dbs2*), and a vacuolar sorting protein (*bro1*). Finally, we identified four mutations in three genes – *crb3*, *grc3* and *mdn1* – coding for subunits of an essential ribosome biogenesis complex that we refer to as the rixosome (*Figure 1D*) and are the focus of this study.

The rixosome contains six unique subunits – three structural subunits Crb3, Rix1 and Ipi1, which form the core of the complex, and three catalytic subunits – the endonuclease Las1, the polynucleotide kinase Grc3 and the AAA-type ATPase Mdn1, which are involved in the processing of ribosomal RNA precursors (*Figure 2A*, *Figure 2—figure supplement 1*; *Fromm et al., 2017*; *Gasse et al., 2015*). All subunits are essential for viability and are conserved from yeast to mammals, including humans (*Figure 2A*). Our screen identified four mutations in three different subunits of the complex – Crb3, Grc3 and Mdn1. We validated three of the mutations by reconstituting them de novo in the original reporter strain. All de novo constructed strains reproduced the mutant phenotypes in plating assays on low adenine medium in the presence and absence of tetracycline (*Figure 2B*). To confirm that the observed maintenance defects were due to loss of heterochromatin, we followed the propagation of the heterochromatin specific histone H3 Lys9 di-methylation (H3K9me2) in the absence of tetracycline and in cultures treated with tetracycline for 10 generations. In the absence of tetracycline, the rixosome mutants showed similar levels of H3K9me2 as the wild-type, which is consistent with the lack of silencing defects under the establishment conditions (*Figure 2C,D*). By contrast, in the tetracycline treated cultures, the levels of H3K9me2 were drastically reduced in the mutants, but not in the wild-type, indicating that loss of silencing of the *10xtetO-ade6$^+$* reporter under the maintenance conditions resulted from loss of the underlying heterochromatic domain (*Figure 2E*).

We next investigated whether the rixosome mutants also cause silencing defects at the endogenous heterochromatic domains. To assess if the mutations affect silencing at the pericentromeric repeats and the mating type locus, we introduced the *crb3-D198N* allele in strains carrying the *ura4$^+$* transgene inserted inside the centromeric *otr1R* and the mating type *cenH* regions, respectively. We then assayed for *ura4$^+$* silencing by plating cells on minimal medium lacking uracil (-Ura) or rich medium containing 5-fluoroacetic acid (5FOA). Uracil supplementation is required for cell growth in the absence of *ura4$^+$* expression, while 5FOA is toxic to cells expressing *ura4$^+$*. Thus, growth on -Ura and 5FOA medium can be used as a readout of *ura4$^+$* expression. In the presence of ongoing establishment from the siRNA generating *otr1R* and *cenH* repeats, silencing at these regions was robust in mutant cells (*Figure 2—figure supplement 2*). Both the mutant and wild-type grew poorly on -Ura medium. At moderate concentrations of 5FOA (0.1%) the growth of the mutant was comparable to that of the wild-type and only at high concentrations (0.2% 5FOA), in which cells are sensitive to very low levels of *ura4$^+$* expression, weak growth defects could be detected (*Figure 2—figure supplement 1*). These results indicate that the *crb3-D198N* mutation has little or no effect on the establishment of heterochromatin at endogenous domains, consistent with the phenotype at the ectopic reporter.

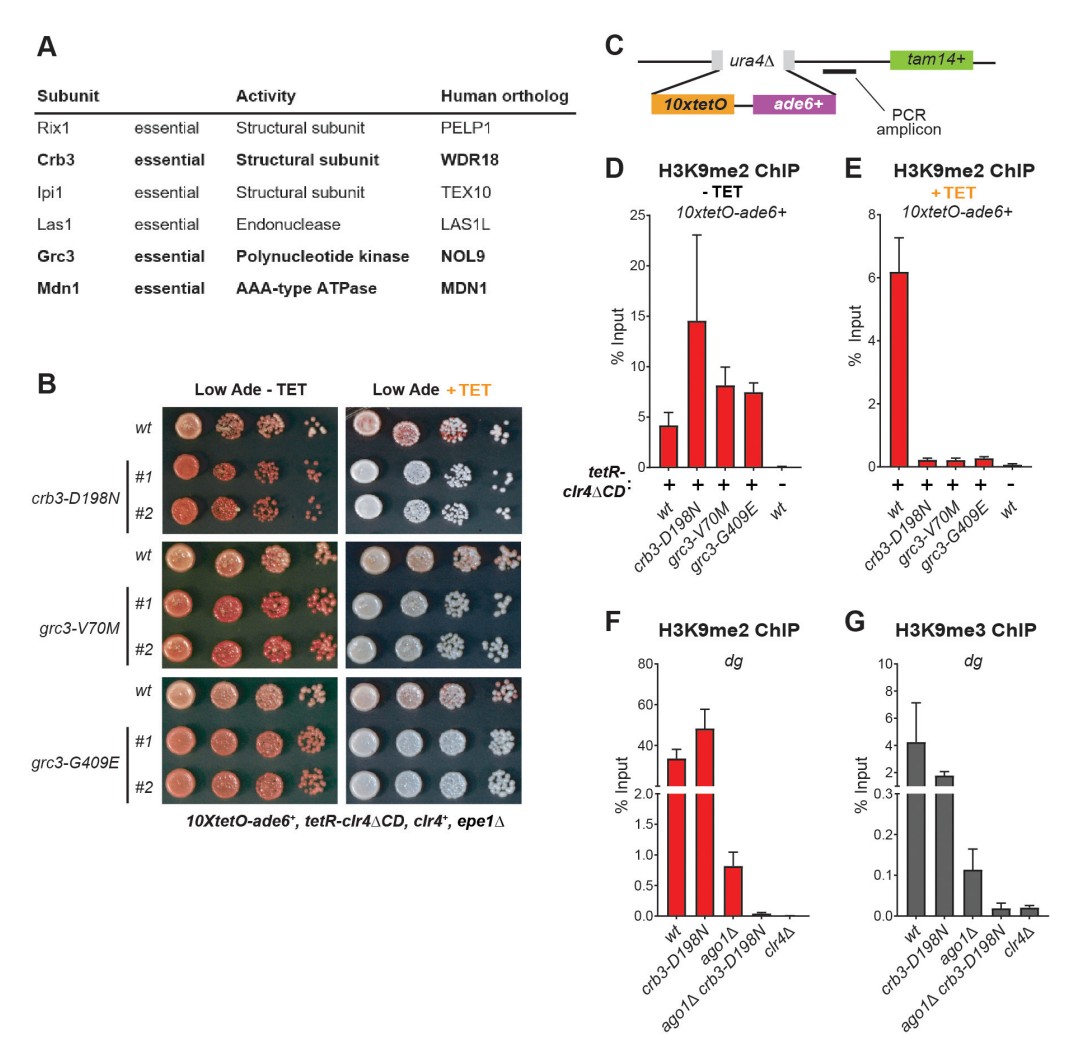

**Figure 2.** The Rixosome is Required for the Inheritance of Heterochromatin. (**A**) Subunits of the rixosome, their specific activities in the context of ribosome biogenesis, and their human orthologs. Subunits identified in our screen are highlighted in bold. (**B**) Silencing assays to validate the phenotypes of mutants derived from the screen. Three mutations in rixosome subunits were reconstituted de novo in *10xtetO-ade6+* reporter cells and plated on low adenine medium in the presence or absence of tetracycline to assess heterochromatin establishment and maintenance. (**C**) Map of the *10xtetO-ade6+* reporter inserted at the endogenous *ura4+* locus. The position of the primer pair used for ChIP-qPCR (PCR amplicon) experiments in panels D and E is indicated. (**D, E**) Chromatin immunoprecipitations (ChIPs) for di-methylated histone H3 Lys 9 (H3K9me2) at the *10xtetO-ade6+* reporter under establishment and maintenance conditions in *epe1Δ* cells, carrying wild-type or mutant rixosome subunits. (**F, G**) ChIP for H3K9me2 and H3K9me3 at the pericentromeric *dg* repeats. Mutations in the rixosome subunit *crb3* abolish H3K9me2 and me3 in the absence of ongoing establishment by the RNAi pathway.

The online version of this article includes the following figure supplement(s) for figure 2:

**Figure supplement 1.** Summary of previously described rixosome functions during ribosome biogenesis (*Gasse et al., 2015*).

**Figure supplement 2.** Heterochromatin establishment at endogenous heterochromatic domains is largely unaffected in rixosome mutations.

**Figure supplement 3.** Effects of the rixosome mutants on rRNA processing A denaturing gel of total RNA, EtBr staining.

Is the rixosome required for the maintenance of H3K9me at the endogenous heterochromatic domains? At the pericentromeric DNA regions, siRNAs from the *dg* and *dh* repeats are required for the establishment of H3K9 methylation and heterochromatic domains (*Volpe et al., 2002*). Mutations in the RNAi pathway, such as deletion of Ago1, remove this active establishment mechanism, resulting in reduced levels of H3K9me2 and me3. (*Sadaie et al., 2004*). The remaining H3K9 methylation levels reflect ongoing heterochromatin maintenance requiring the Clr4 read-write mechanism (*Ragunathan et al., 2015*). To test if the rixosome is required for maintenance of the remaining

H3K9 methylation, we introduced the *crb3-D198N* mutation in the *ago1⁺* and *ago1Δ* cells and compared the levels of H3K9me2 and me3 at the pericentromeric *dg* repeats to the levels observed in *clr4Δ* cells, which lack histone H3K9 methylation. In *ago1⁺* cells, which have continuous establishment, the *crb3-D198N* mutant did not significantly reduce the levels of H3K9me2 (**Figure 2F**), consistent with the results in the *10xtetO-ade6⁺* reporter locus under establishment conditions (**Figure 2B,D**). The *crb3-D198N* mutant caused a 2-fold reduction in the levels of H3K9me3, which may indicate a role for the rixosome in heterochromatin maturation (**Figure 2G**). By contrast, in *ago1Δ* cells, where establishment is abolished, H3K9me2 and me3 enrichment was reduced to background levels by the *crb3-D198N* mutation (**Figure 2F,G**). These results show that while the establishment of RNAi-dependent H3K9me and silencing at endogenous pericentromeric repeats does not require the rixosome, the epigenetic maintenance of H3K9me at these regions is completely rixosome-dependent.

Since the rixosome plays an essential role in ribosome biogenesis, we tested whether the rixosome mutations isolated in our screen affect rRNA processing. rRNA precursors accumulate in mutants which disrupt processing and can be detected by gel electrophoresis. In a known rRNA processing mutant in the Crb3 subunit of the rixosome (**Kitano et al., 2011**), rRNA precursors accumulate at elevated temperatures (**Figure 2—figure supplement 3**). We observed weak rRNA processing defects at elevated temperatures in cells carrying *crb3-D198N* and *grc3-G409E* mutations. By contrast, no rRNA precursors accumulated in the *grc3-V70M* mutant (**Figure 2—figure supplement 3**), which we therefore conclude is a separation-of-function allele best suited for further analysis of rixosome function in heterochromatin formation.

## A mutation that abolishes rixosome heterochromatic localization

To determine whether the rixosome mutations affect the integrity of the complex or its interactions with other proteins, we affinity purified wild-type and mutant rixosome complexes and analyzed them by quantitative tandem mass tag mass spectrometry (TMT-MS) (**Navarrete-Perea et al., 2018**). The rixosome co-purifies with native heterochromatin fragments (**Iglesias et al., 2020**). We therefore used this native purification protocol to test whether the rixosome mutations affect its interaction with heterochromatin (**Iglesias et al., 2020**). We tagged the C-terminus of the Crb3 subunit with a tandem affinity purification (TAP) tag (**Puig et al., 2001**), which did not interfere with the maintenance of silencing at the *10xtetO-ade6⁺* reporter (**Figure 3—figure supplement 1A**). We then purified the rixosome using its Crb3-TAP subunit from wild-type, Crb3-D198N, or Grc3-V70M cell extracts (**Figure 3A**, **Figure 3—figure supplement 1B,C**) and analyzed the purifications by TMT-MS. The Crb3-TAP purification from wild-type cells pulled down the core subunits reported previously to be part of the rixosome (**Figure 3B**, **Figure 3—figure supplement 1D**) and its rRNA processing and ribosome biogenesis interaction network (**Supplementary files 3** and **4** Tables S3 and S4). Importantly, we also recovered the heterochromatin-associated protein Swi6, which was previously shown to associate with the rixosome in an H3K9me-independent manner (**Iglesias et al., 2020**). When we purified Crb3-TAP from cells carrying the *crb3-D198N* mutation, we detected a modest reduction in the amount of co-precipitated Swi6 in the mutant (**Figure 3—figure supplement 1D**), which was not explained by a decrease in the overall Swi6 protein levels (**Figure 3—figure supplement 1E**). However, other interactions, including those between the subunits of the rixosome complex (**Figure 3—figure supplement 1D**, **Supplementary file 4**), were also weakly affected, indicating that the stability of the complex in the *crb3-D198* mutant may have been compromised. These results were consistent with the mild rRNA processing defects observed in this mutant at elevated temperatures (**Figure 2—figure supplement 3**). By contrast, when we purified Crb3-TAP from cells carrying the *grc3-V70M* mutation, the levels of Swi6 copurifying with the mutant complex were dramatically reduced, but the mutation had little or no effect on the overall integrity of the rixosome and its other interactions (**Figure 3B**, **Supplementary file 3**). Furthermore, neither the *crb3-D198N*, nor the *grc3-V70M* mutations had any effect on the total levels of Swi6 in the cell, ruling out that the loss of association of the rixosome with Swi6 could be an indirect effect of reduced Swi6 expression or stability (**Figure 3C**, **Figure 3—figure supplement 1E**). These results demonstrate that the Grc3-V70M mutation specifically disrupts the association of the rixosome complex with Swi6 and further support a specific role for the complex in heterochromatin maintenance, independent of its essential function in ribosome biogenesis. The Grc3-V70M mutation is located in the unstructured N-terminus of Grc3 within a VxVxV sequence motif, a variant of the consensus

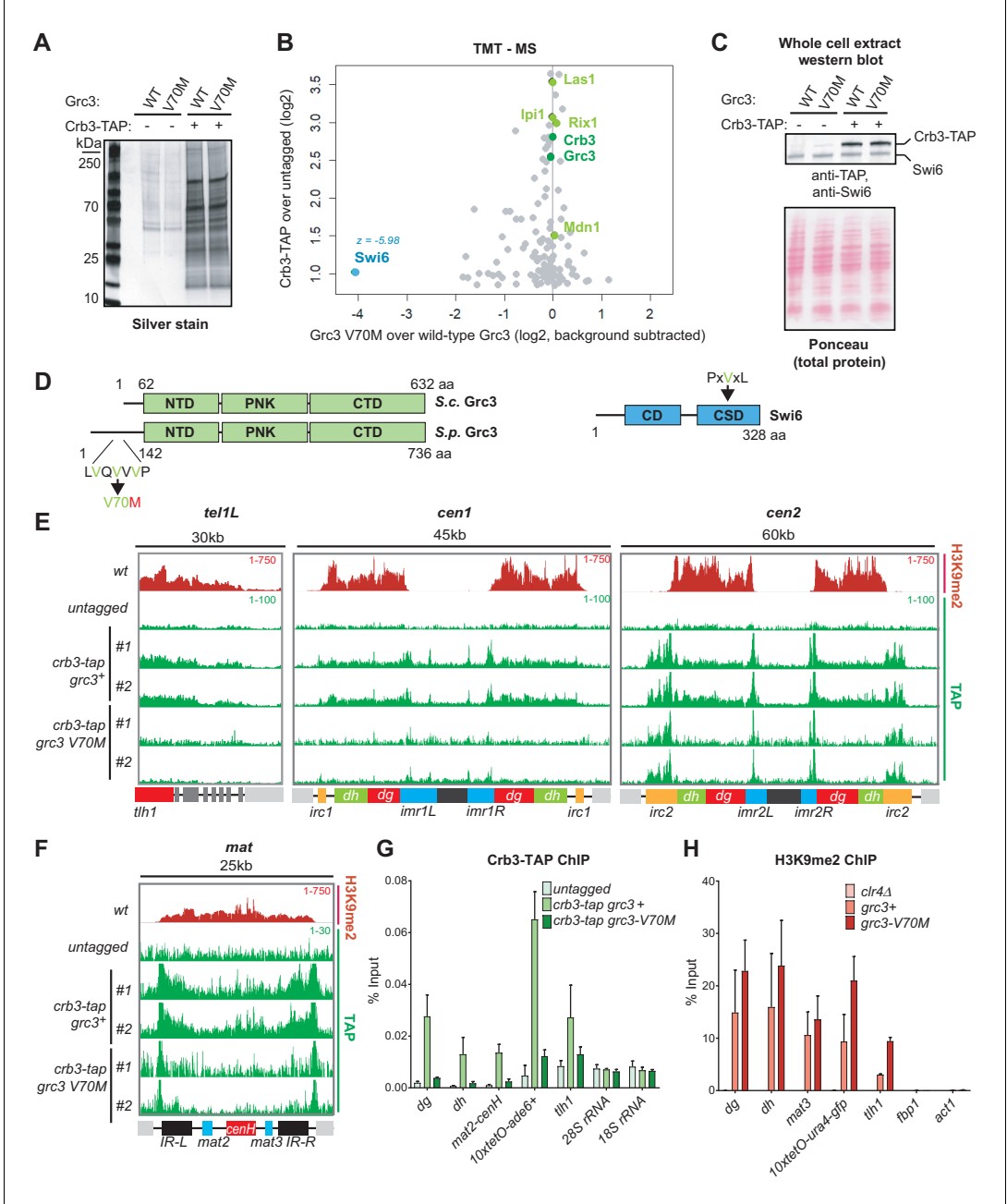

**Figure 3.** A mutation in the rixosome Grc3 subunit abolishes its interaction with the HP1 protein Swi6 and releases the mutant complex from heterochromatin. (**A**) Silver staining of immunoprecipitations of wild-type and *grc3-V70M* mutant rixosome complexes from cell extracts. (**B**) Tandem mass tag mass spectrometry (TMT-MS) of the immunoprecipitations in A. Subunits of rixosome highlighted in green, HP1 protein Swi6 in blue, ribosome biogenesis and other factors in gray. For detailed description of TMT-MS data analysis see the STAR Methods section. (**C**) Top, western blot from whole cell extracts for the rixosome subunit Crb3-TAP and the HP1 protein Swi6 from *grc3+* and *grc3-V70M* cells. Bottom, ponceau staining is shown as a loading control. (**D**) Schematics of domain organizations of Grc3 (left) and Swi6 (right). Homologous regions of *S. pombe* (S.p.) Grc3 and *S. cerevisiae* (S.c.) Grc3 are indicated (*Pillon et al., 2017*). The extended N-terminus of S.p. Grc3 lacks similarity to *S.c.* Grc3. The position and amino acid context of the V70M mutation are shown. The three residues highlighted in green are a variant of a consensus motif (PxVxL), recognized by the chromoshadow domain of HP1 proteins. NTD, N-terminal domain; PNK, polynucleotide kinase domain; CTD, C-terminal domain; CD, chromodomain; CSD, chromoshadow domain. (**E, F**) ChIP sequencing profiles of H3K9me2 and Crb3-TAP at heterochromatin in *grc3+* and *grc3-V70M* cells. *tel*, telomere; *cen*, centromere; *mat*, mating type locus; kb, kilobases. Numbers in top right corner of each panel show reads per million. For detailed notations, see *Figure 2—figure supplement 1* legend. (**G, H**) ChIP-qPCR quantification of rixosome and H3K9me2 enrichment at different heterochromatic regions in wild-type and *grc3-V70M* mutant *epe1Δ* cells. *dg* and *dh*, tandem repeats in pericentromeric DNA repeats; *tlh1*, subtelomeric gene; *act1* and *fbp1*, euchromatic controls.

*Figure 3 continued on next page*

*Figure 3 continued*

The online version of this article includes the following figure supplement(s) for figure 3:

**Figure supplement 1.** Identification of proteins associated with wild-type and *crb3-D198N* mutant complexes.

**Figure supplement 2.** Recruitment of rixosome complexes to heterochromatin in wild-type, *crb3-D198N* and *grc3-V70M* cells.

PxVxL motif of proteins, which interact directly with the chromoshadow domain of HP1 proteins (*Figure 3D*; *Thiru et al., 2004*). The substitution of the central valine in this motif, which is crucial for the interaction between HP1 and HP1-interacting proteins (*Thiru et al., 2004*), with methionine, would be expected to disrupt the binding surface. Therefore, Swi6 likely interacts directly with the N-terminus of Grc3 via its chromoshadow domain.

We next asked whether the Grc3-V70M and Crb3-D198N mutations affect the localization of the complex to heterochromatic DNA regions. We performed chromatin immunoprecipitations (ChIPs) of Crb3-TAP from wild-type, *crb3-D198N* and *grc3-V70M* cells, followed by sequencing or qPCR. As shown in *Figure 3E,F and G*, Crb3-TAP co-localized with H3K9me2-containing heterochromatic domains. In addition, Crb3-TAP localized to the clusters of tRNA genes concentrated at the boundaries of the centromeres and to the inverted repeat regions, *IR-L* and *IR-R*, of the *mat* locus (*Figure 3E,F*), which carry B-box sites and recruit TFIIIC (*Noma et al., 2006*). In the *crb3-D198N* mutant, which only partially reduced the association of the complex with the heterochromatic protein Swi6, the localization of the rixosome was only modestly reduced (*Figure 3—figure supplement 2A*). This result suggests that the impairment of epigenetic maintenance in *crb3-D198N* mutant cells results from a rixosome defect downstream of its localization to heterochromatic domains. By contrast, in the *grc3-V70M* mutant, heterochromatin localization of Crb3-TAP was reduced to background levels (*Figure 3E,F and G*, *Figure 3—figure supplement 2B*). The *grc3-V70M* mutation did not affect the localization of Crb3-TAP to the boundaries of the centromeres or the mating type locus, indicating that the recruitment of the rixosome to these regions is independent of its ability to interact with Swi6 (*Figure 3E,F*). Furthermore, Crb3-TAP localized to the ectopically induced domain of H3K9 methylation at the *tetO-ade6+* reporter and this localization was abolished in *grc3-V70M* cells (*Figure 3—figure supplement 2C*), demonstrating that heterochromatin formation was sufficient for rixosome recruitment. Finally, Crb3-TAP was not enriched at the ribosomal DNA loci on chromosome III (25S and 18S rRNA) or the subtelomeric *tel1R and tel2L* regions in wild-type cells, despite the presence of H3K9me2 (*Figure 3—figure supplement 2D*). In addition to heterochromatic regions, the rixosome showed enrichment at a number of euchromatic loci, which did not contain H3K9me2 and were not affected by the *grc3-V70M* mutant (*Figure 3E,F*, *Figure 3—figure supplement 2E*), including protein coding, tRNA and 5S rRNA genes. Consistent with unperturbed establishment of native heterochromatin in *grc3-V70M* mutant cells (*Figure 1D*, *Figure 2B*), the mutation did not affect the levels of H3K9me2 at various heterochromatic loci (*Figure 3H*). These results suggest that the failure of rixosome complexes, carrying the Grc3-V70M mutant subunit, to localize to heterochromatin results in defective epigenetic maintenance.

## The rixosome promotes Dhp1/XRN2-mediated RNA degradation

In the context of rRNA processing in *S. cerevisiae*, the endonuclease and polynucleotide kinase activities of the rixosome cleave the rRNA precursor and prepare the resulting RNA ends for downstream processing (*Figure 2—figure supplement 1*; *Castle et al., 2012*; *Castle et al., 2013*; *Schillewaert et al., 2012*; *Gasse et al., 2015*). The processing of the 3′ end is carried out by the exosome, in a molecular hand-off event, where the exosome subunit Dis3 removes the 2′−3′ cyclic phosphate produced by the Las1 endonucleolytic cleavage and hands off the 3′end to the Rrp6 subunit for further degradation (*Fromm et al., 2017*). The processing of the 5′ end is carried out by the nuclear 5′−3′ exonuclease Dhp1, a homolog of the budding yeast Rat1 and human XRN2 proteins, which recognizes the free 5′ phosphate group deposited by Grc3 (*Gasse et al., 2015*). Moreover, in *S. pombe*, Dhp1 has previously been shown to promote transcription termination within heterochromatin and a *dhp1* mutation impairs the silencing of some heterochromatic reporter genes, although how heterochromatic RNAs become substrates for Dhp1 had remained unknown (*Chalamcharla et al., 2015*; *Tucker et al., 2016*). We hypothesized that the rixosome is specifically recruited to heterochromatin to initiate the degradation of heterochromatic RNAs, by channeling

them through the 5′−3′ and 3′−5′ exonucleases Dhp1 and Dis3/Rrp6, respectively. To test this hypothesis, we introduced the previously described *dhp1-1* (*Shobuike et al., 2001*), and *dis3-54* (*Murakami et al., 2007*; *Ohkura et al., 1988*) mutant alleles into cells carrying the *10xtetO-ade6⁺* reporter. Under establishment conditions, in the absence of tetracycline, the *dhp1-1* allele had no effect on silencing of the *10xtetO-ade6⁺* gene, but abolished silencing under maintenance conditions, in the presence of tetracycline, closely phenocopying the rixosome mutants (*Figure 4A*). Consistent with the silencing results, chromatin immunoprecipitation showed that H3K9me2 was lost in *dhp1-1* cells, in the presence of tetracycline but not in its absence (*Figure 4B*). Relative to the *dhp1-1* allele, the *dis3-54* allele disrupted maintenance to a lesser extent (*Figure 4—figure supplement 1*). Together, these results suggest that 5′−3′ mediated RNA degradation by Dhp1, rather than 3′−5′ mediated RNA degradation by the exosome, is the key event downstream of the rixosome required for epigenetic inheritance.

We next asked whether a catalytic activity of the rixosome, which is required for RNA degradation, is also required for its role in the inheritance of gene silencing. We inserted either wild-type or catalytically dead Grc3, under the control of the native *grc3* promoter, at the chromosomal *leu1* locus of *grc3-G409E* (*emd13*) mutant cells, which are maintenance defective. The catalytically dead mutant carried substitutions of two conserved catalytic site residues (K252A, S253A), which have

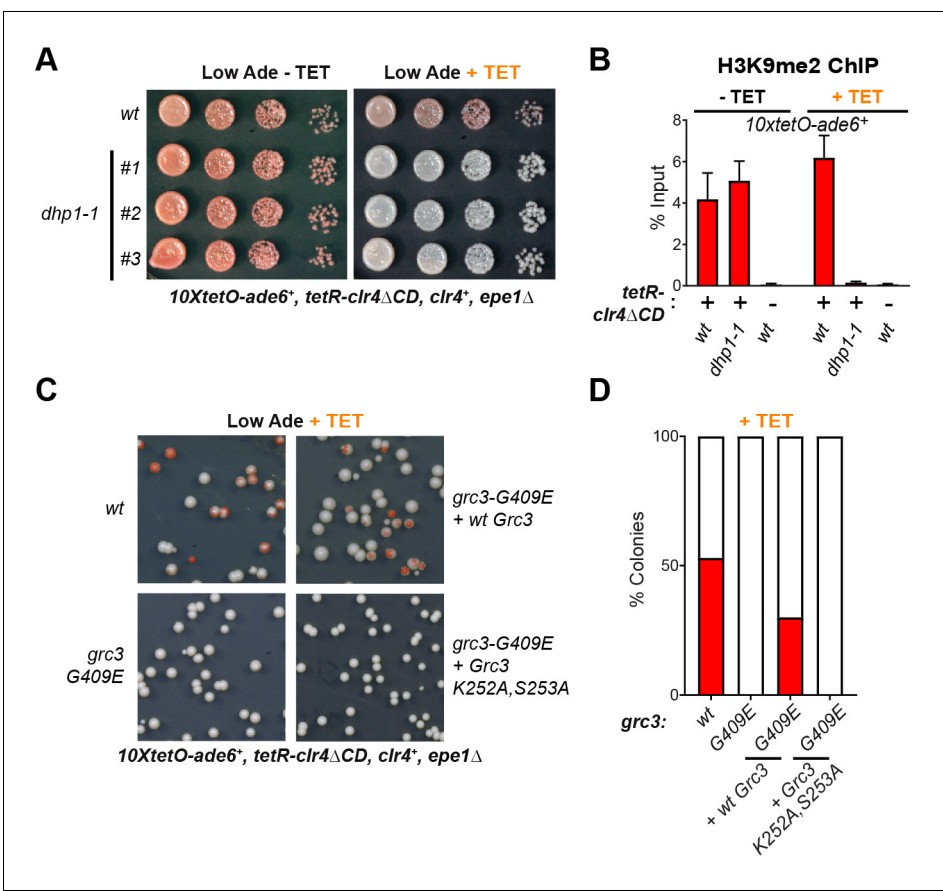

**Figure 4.** The 5′−3′ exonuclease Dhp1/Xrn2 and the catalytic activity of the rixosome subunit Grc3 are essential for epigenetic inheritance of gene silencing. (**A**) Silencing assay for heterochromatin establishment and maintenance at the *10xtetO-ade6+* reporter locus in a Dhp1/Xrn2 hypomorphic mutant, *dhp1-1*. (**B**) ChIP for H3K9me2 at the *10xtetO-ade6⁺* reporter locus under establishment and maintenance conditions in cells carrying *dhp1⁺* or *dhp1-1* alleles. For primer positions, see *Figure 2C*. (**C**) Rescue of maintenance defect by expression of wild-type or catalytically dead (K252A, S253A) Grc3 in *grc3-G409E* cells. Silencing assay on low adenine medium shows that catalytic activity is essential to restore maintenance.(**D**) Quantification of the plating assays in (**C**). The online version of this article includes the following figure supplement(s) for figure 4:

**Figure supplement 1.** A mutation that impairs exosome function does not affect heterochromatin maintenance.

been shown to be crucial for the kinase activity of Grc3 in *S. cerevisiae* (*Braglia et al., 2010*). To assess rescue of the maintenance defect of *grc3-G409E* cells by either wild-type or catalytically dead Grc3, we plated cells on low adenine medium in the presence of tetracycline. Expression of wild-type Grc3 largely restored the maintenance defect of the *grc3-G409E* mutant cells (*Figure 4C,D*). By contrast, expression of catalytically dead Grc3-K252A, S253A did not rescue the maintenance defect (*Figure 4C,D*). These observations demonstrate that the kinase activity of Grc3, which prepares RNA substrates for 5′−3′ exonucleolytic degradation by Dhp1, is essential for epigenetic inheritance of heterochromatin.

The above results raise questions about how rixosome-dependent RNA degradation may specifically contribute to heterochromatin maintenance. We hypothesized that the complex may be required for the removal of heterochromatic RNAs that may not interfere with silencing during continuous establishment but may need to be degraded for successful epigenetic maintenance. To test the effects of the mutations on the steady-state levels of RNAs transcribed from the *10xtetO* reporter locus, we used the previously described *10xtetO-ura4⁺-gfp* version of the reporter (*Ragunathan et al., 2015*). Unlike *ade6*, which is present in the genome in two copies, an ectopic wild-type copy and endogenous copy carrying a point mutation, the *gfp* transgene presents a unique sequence for quantitative reverse transcription (qRT) and RNA Pol II ChIP. Under establishment condition, we did not observe an increase in *gfp* reporter RNA levels in the *grc3-V70M* mutant relative to wild-type cells. We reasoned that other RNA degradation pathways may act redundantly to mask the effects of the rixosome on RNA levels under strong establishment conditions at the reporter locus. We deleted a component of the nuclear TRAMP complex (*cid14Δ*) and a component of the Ccr4-Not complex (*caf1Δ*) (*LaCava et al., 2005*; *Bühler et al., 2007*; *Tucker et al., 2001*; *Brönner et al., 2017*), the other major RNA degradation pathways previously shown to impact heterochromatin (*Figure 5A*). TRAMP channels nuclear RNAs for degradation to the exosome, while the Ccr4-Not complex is the principal deadenylase, which prepares transcripts for decapping and degradation in the cytoplasm by the exosome and Xrn1 (*Tucker et al., 2001*; *LaCava et al., 2005*). Deletion of *cid14⁺* did not affect heterochromatin maintenance and the *cid14Δ grc3-V70M* double mutant cells did not show any increase in *gfp* RNA relative to *cid14Δ* alone (*Figure 5—figure supplement 1A,B*). By contrast, the *caf1Δ grc3-V70M* double mutant showed a marked accumulation of *gfp* transcripts relative to *caf1Δ* alone (*Figure 5B and C*). The increase in RNA levels in the double mutant was not accompanied by a small increase in RNA Pol II occupancy (*Figure 5D,E*). We conclude that even under strong establishment conditions, in which the *10xtetO-ura4⁺-gfp* reporter is strongly silenced at the phenotypic level, some RNA is transcribed from the locus and is targeted for degradation by the rixosome and Ccr4-Not complex, and that degradation of such 'toxic' RNA by heterochromatin-associated rixosome is required for epigenetic inheritance of H3K9me.

As shown in *Figure 2E,F* and *Figure 2—figure supplement 1*, rixosome mutants have only slight defects in pericentromeric silencing, indicating that the rixosome is not required for RNAi-mediated heterochromatin establishment. Nevertheless, we reasoned that since the rixosome can localize to pericentromeric repeats through association with Swi6 (*Figure 3E*), it might participate in degradation of RNAs transcribed from the repeats in a parallel and redundant manner alongside RNAi. If this were the case, we would expect an increase in the level of siRNAs in the absence of the rixosome as more RNA would be available for processing by the RNAi machinery (*Figure 5F*). We assessed the levels of siRNA generated from the pericentromeric *dg* repeats in wild-type, *grc3-V70M,* and *dhp1-1* cells by northern blotting and, consistent with the above hypothesis, found that both mutations caused a marked increase in siRNAs accumulation (*Figure 5G, lanes 1–3 and H*; *Figure 5—figure supplement 1C–E*). Moreover, the *grc3-V70M dhp1-1* double mutant had siRNA levels comparable to those of each of the single mutants, suggesting that the rixosome and the exonuclease Dhp1 act in the same pathway (*Figure 5G*, lane 4, *Figure 5—figure supplement 1C,D,E*). Consistent with the requirement for Swi6 in recruitment of the rixosome to heterochromatin (*Iglesias et al., 2020*), *dg* siRNA levels were similar between *swi6Δ grc3⁺* and *swi6Δ grc3-V70M* cells, indicating that the ability of the rixosome pathway to compete with RNAi required its Swi6-mediated recruitment (*Figure 5G, lanes 5 and 6, and H*; *Figure 5—figure supplement 1C,D,E*). Swi6 stabilizes the RNAi machinery on heterochromatin and in its absence centromeric siRNA levels are reduced (*Motamedi et al., 2004*). This provides an explanation for the reduced siRNA levels in both *swi6Δ* single mutant and *swi6Δ grc3-V70M* double mutant cells relative to *grc3-V70M* and *dhp1-1* mutant cells (*Figure 5G, lanes 5 and 6, and H*). The increased accumulation of pericentromeric *dg* siRNAs in the above mutant cells

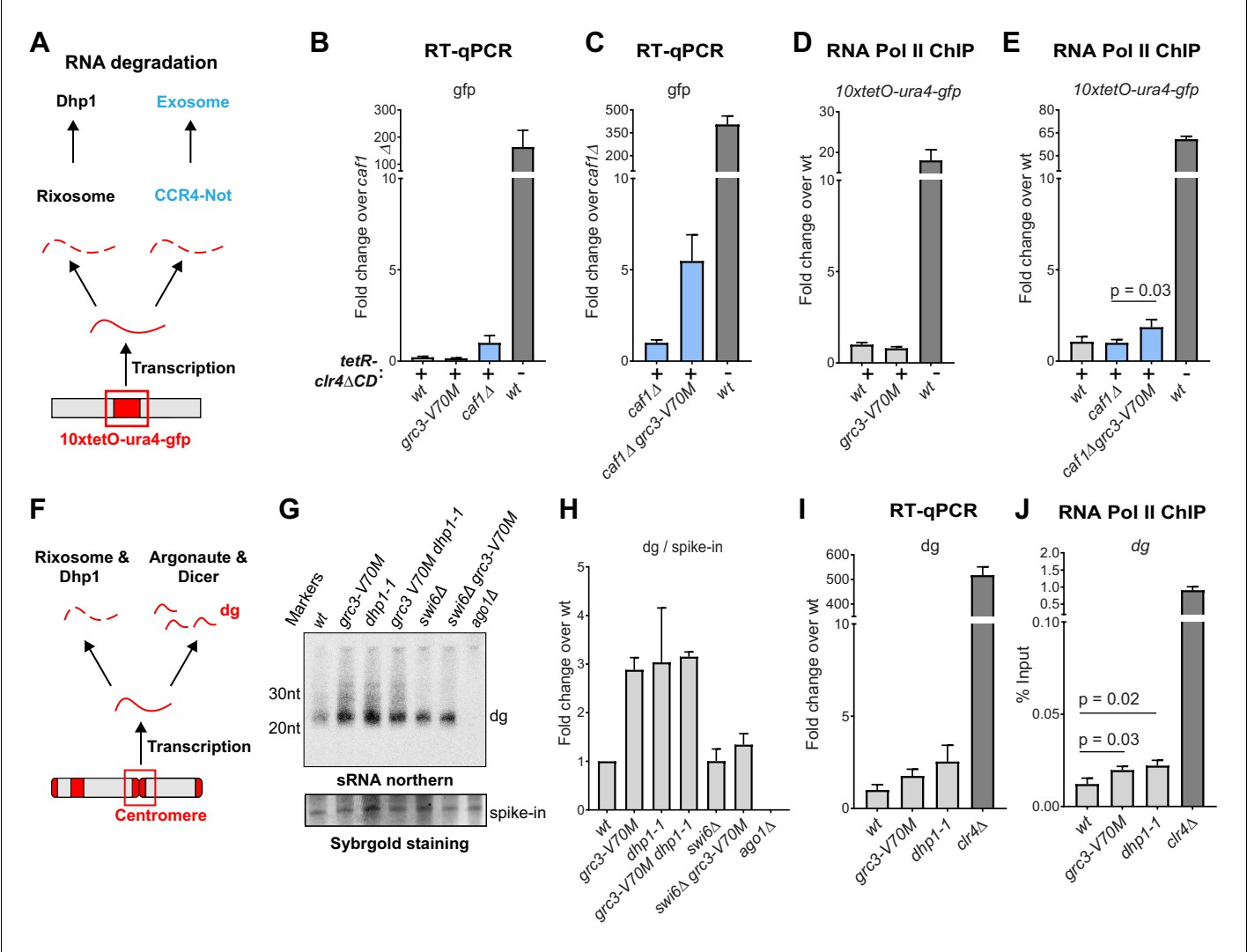

**Figure 5.** The rixosome channels heterochromatic RNAs for degradation via the 5′—3′ exonuclease Dhp1/Xrn2. (**A**) Diagram of possible RNA degradation pathways at the *10xtetO-ura4-gfp/ade6*⁺ reporter. The rixosome cleaves RNAs produced from the reporter on heterochromatin and targets them for degradation by the nuclear Dhp1/Xrn2 exonuclease. If an RNA escapes degradation by the rixosome, it is deadenylated by the Ccr4-Not complex, in which Caf1 is a key subunit, in preparation for decapping and degradation by the cytoplasmic exosome. Deletion of the deadenylation pathway unmasks the effect of the rixosome and Xrn2/Dhp1 on reporter RNAs. (**B, C**) Reverse transcription (RT) -qPCR assay for RNA accumulation at the *10xtetO-ura4-gfp/ade6*⁺ reporter locus in wild-type, *grc3-V70M* and *caf1Δ* cells (**B**) and in *caf1Δ and caf1Δ grc3* cells (**C**). All strains are also *epe1Δ*. Caf1 is a subunit of the major cytoplasmic RNA deadenylase complex, Ccr4-Not. (**D, E**) ChIP for RNA polymerase II (Pol II) at the reporter locus in wild-type and *grc3-V70M* cells (**D**) and in wild-type, *caf1Δ and caf1Δ grc3-V70M* cells (**E**), also carrying *epe1Δ*. (**F**) Model for the role of the rixosome in RNA degradation at the pericentromeric repeats. The rixosome acts on full length RNAs transcribed from the repeats in competition with RNAi and targets them for degradation through the exonuclease Xrn2/Dhp1. When the rixosome is released from heterochromatin or when Dhp1 activity is reduced, *dg* transcripts normally degraded by the rixosome and Dhp1 are channeled for processing by RNAi, resulting in increased siRNA accumulation. (**G**) Small RNA Northern blot, measuring the accumulation of small RNAs originating from the pericentromeric *dg* repeats in wild-type and different mutants. A 20mer oligoribonucleotide was added to each sample prior to purification as an internal loading control and used for normalization. Fold changes with standard deviations relative to the spike-in 20mer are shown below each lane. See *Figure 5—figure supplement 1E* for uncropped images. (**H**) Quantification of the blot from panel G for *dg* siRNAs, normalized to the internal spike-in control. The plot displays two data points, their mean and standard deviation for each genotype. See *Figure 5—figure supplement 1D* for normalization using snoR69. (**I**) ChIP for RNA polymerase II (Pol II) at the pericentromeric *dg* repeats.

The online version of this article includes the following figure supplement(s) for figure 5:

**Figure supplement 1.** Extended data for RNA degradation at heterochromatin.

was accompanied by only a modest increase in full-length *dg* transcripts (*Figure 5I*). These results are consistent with the hypothesis that excess *dg* transcripts, resulting from loss of rixosome-mediated degradation, are rapidly processed into siRNAs by the RNAi pathway (*Figure 5F*). Finally, RNA polymerase II (RNA Pol II) occupancy at the *dg* repeats was only weakly affected (*Figure 5J*), suggesting that the mutations did not affect the levels of transcription from the repeats. We therefore conclude that heterochromatin-associated rixosome cleaves and phosphorylates target RNAs so that they become substrates for degradation by the Dhp1 exonuclease.

## Role for the rixosome in heterochromatin spreading

To investigate whether mutations in the rixosome or Dhp1 affect the stability of heterochromatic domains genome-wide, we performed ChIP for H3K9me2 and me3, followed by sequencing or qPCR, in wild-type and mutant cells. While the pattern or levels of H3K9me2 were generally not perturbed in *grc3-V70M* and *dhp1-1* mutants relative to wild-type cells, the mutant cells displayed a modest reduction in H3K9me3 levels. This reduction only occurred at heterochromatic domains where the rixosome was enriched: the *mat* locus (*Figure 6—figure supplement 1A and B*), the pericentromeric repeats of *cen1*, *cen2* and *cen3* (*Figure 6—figure supplement 1A,C,D*) and the *10xtetO-ade6⁺* reporter under establishment conditions (*Figure 6—figure supplement 1A*). Since the read-write function of Clr4 is required for the transition from H3K9me2 to me3 (*Jih et al., 2017*), these results raised the possibility that the rixosome-Dhp1 pathway was required for efficient Clr4 read-write.

In addition to efficient H3K9me3 deposition and epigenetic inheritance, Clr4 read-write is required for RNAi-independent spreading of H3K9me away from nucleation sites (*Zhang et al., 2008*; *Jih et al., 2017*). We therefore tested the hypothesis that spreading of heterochromatin at the *L(BglII):ade6⁺* transgene (*Figure 6A*; *Ayoub et al., 1999*) at the mating type locus may depend on rixosome- and Dhp1-dependent RNA degradation. In wild-type cells, heterochromatin spreads from the RNAi initiating centromere Homology (*cenH*) region into the *mat2p* gene and the adjacent *IR-L* boundary region (*Zhang et al., 2008*). Transgenes inserted into the *IR-L* region display variable levels of silencing due to stochastic spreading of H3K9 methylation and can therefore be used as a readout for heterochromatin spreading. As a control, we first tested the effects of the read-write deficient *clr4-W31G* chromodomain mutant on *L(BglII):ade6⁺* silencing. As expected, the *L(BglII):ade6⁺* transgene became fully derepressed in *clr4-W31G* cells, indicating that Clr4 read-write was essential for the spread of silencing away from the *cenH* nucleation region (*Figure 6B*). We then replaced the chromosomal copies of the *grc3⁺* and *dhp1⁺* genes with the *grc3-V70M* and *dhp1-1* mutant alleles in cells carrying the transgene. Both mutants abolished the silencing of the *ade6⁺* transgene, in a manner similar to the *clr4-W31G* mutant, suggesting that the rixosome and Dhp1 were required for the spread of silencing into the transgene (*Figure 6C,D*). Consistent with the colony color assays, analysis of RNA levels using qRT-PCR showed strong accumulation of *ade6⁺* RNA in the *grc3-V70M* and *dhp1-1* mutant cells relative to wild-type cells, to an equal or greater extent as that observed in *clr4-W31G* and *clr4Δ* cells, indicating that *L(BglII):ade6⁺* transgene silencing was completely rixosome- and Dhp1-dependent (*Figure 6E*, primer pair 2). As expected, silencing of the *mat2P* gene, which is H3K9me-independent (*Hansen et al., 2011*), was not affected by any of the above mutations (*Figure 6E*, primer pair 4). Moreover, ChIP-qPCR experiments showed that *grc3-V70M*, *dhp1-1*, and *clr4* mutations (*clr4-W31G* and *clr4Δ*) did not significantly affect RNA Pol II occupancy at the *cenH* region and the *ade6⁺* transgene (*Figure 6F*), suggesting that silencing occurred primarily at the level of RNA degradation. The loss of *L(BglII):ade6⁺* transgene silencing in *grc3-V70M* and *dhp1-1* cells was accompanied by impaired spreading of H3K9me2 and H3K9me3 from the *cenH* region into the *ade6⁺* transgene and leftward into the *IR-L* boundary, while spreading into the *mat2P* gene occurred with similar efficiency in mutant and wild-type cells (*Figure 6G and H*). Moreover, ChIP-seq experiments showed that when the *ade6⁺* reporter was absent, the *grc3-V70M* and *dhp1-1* mutations had little or no effect on the spreading of H3K9me into the native *IR-L* region, supporting the idea that the rixosome was required for spreading of H3K9me across the transcriptionally active *ade6⁺* transgene (*Figure 6—figure supplement 1B*). Consistent with previous results (*Zhang et al., 2008*), in *clr4-W31G* cells, the levels of H3K9me2 and H3K9me3 were strongly reduced and did not spread outside the *cenH* region (*Figure 6G and H*). These results uncover an essential role for the rixosome in spreading of H3K9me and silencing over a transcription unit that is inserted distal to a heterochromatin nucleation center.

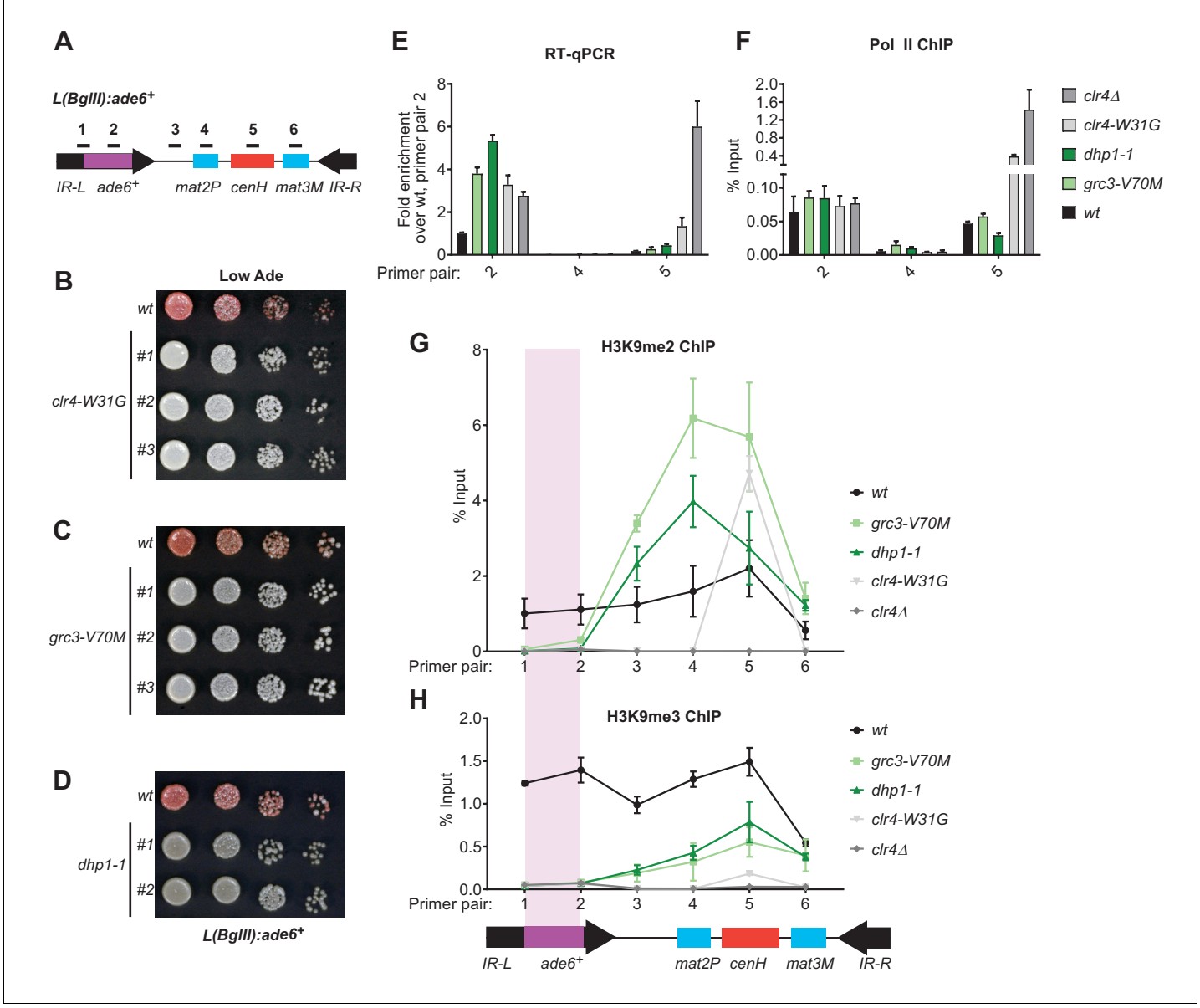

**Figure 6.** Rixosome and Dhp1 are required for RNAi-independent heterochromatin spreading and promote the Clr4-catalyzed transition H3K9me2 to H3K9me3. (**A**) Schematic of DNA sequence organization of the mating type locus with position of the *L(BglII):ade6+* reporter and centromere homology (*cenH*, red) region highlighted. For detailed notations, see *Figure 2—figure supplement 1* legend. (**B, C, D**) Silencing assays of the *L(BglII):ade6+* reporter in wild type, *clr4-W31G*, *grc3-V70M* and *dhp1-1* cells. (**E**) Qunatitative Reverse Transcription PCR (qRT) for transcripts along the mating type locus in wild type cells and cells carrying *clr4-W31G*, *grc3-V70M*, *dhp1-1*, and *clr4Δ* mutant alleles. (**F**) ChIP-qPCR for RNA Polymerase II occupancy along the mating type locus in wild type cells and cells carrying *clr4-W31G*, *grc3-V70M*, *dhp1-1*, and *clr4Δ* mutant alleles. (**G, H**) ChIP-qPCR for H3K9me2 and H3K9me3 at the *L(BglII):ade6+* reporter and surrounding regions at the mating type locus in wild type cells and cells carrying *clr4-W31G*, *grc3-V70M*, *dhp1-1*, and *clr4Δ* mutant alleles.

The online version of this article includes the following figure supplement(s) for figure 6:

**Figure supplement 1.** The rixosome and Dhp1 are required for efficient H3K9 tri-methylation.

## Discussion

Our ability to perform a sensitive genetic screen for mutations that specifically abolish heterochromatin maintenance allowed us to isolate separation-of-function alleles and demonstrate a vital role for the essential ribosome biogenesis complex, the rixosome, in epigenetic inheritance of heterochromatin and spreading of H3K9me along the chromatin fiber. The rixosome is recruited to

heterochromatin by the HP1 protein, Swi6, to promote the degradation of RNAs transcribed by RNA pol II within heterochromatic domains. In the context of heterochromatin, the rixosome may perform a function that is analogous to that of the cleavage and polyadenylation specificity factor (CPSF) complex at euchromatic mRNAs or the integrator complex which has been shown to release paused RNA pol II complexes (*Shi and Manley, 2015*, *Elrod et al., 2019*). However, unlike CPSF, which is recruited by specific polyadenylation signals in the 3' ends of mRNAs, the rixosome is recruited throughout heterochromatic domains via association with Swi6/HP1 and targets heterochromatic RNAs for degradation rather than polyadenylation and export. The role for the rixosome in heterochromatic RNA degradation is supported by our demonstration that (1) heterochromatic RNAs accumulate in cells carrying rixosome mutations, (2) the kinase activity of the Grc3 subunit of the rixosome, which prepares RNA for degradation by the 5'−3' exoribonuclease Dhp1/XRN2, is required for its heterochromatin maintenance function, and (3) a mutation in Dhp1 has the same heterochromatin formation defects as rixosome mutants. The conservation of all rixosome subunits from yeast to human raises the possibility that it functions in chromatin silencing in other eukaryotes.

Our findings suggest a new role for RNA degradation, promoted by recruitment of the rixosome, in formation and epigenetic maintenance of heterochromatin. The rixosome contains subunits with multiple catalytic activities (*Castle et al., 2012*; *Schillewaert et al., 2012*; *Castle et al., 2013*; *Gasse et al., 2015*; *Fromm et al., 2017*; *Figure 2—figure supplement 1*). These activities, which take place on the pre-60S ribosomal particles, include the Las1-mediated endonucleolytic cleavage of the internal transcribed spacer 2 (ITS2) in 27S pe-rRNA, followed by phosphorylation of the 5'-OH cleavage product to generate a 5'-$PO_4$ end, which becomes a substrate for trimming by the 5'−3' exonuclease Dhp1/XRN2 leading to formation of mature 25S rRNA (*Figure 2—figure supplement 1*; *Fromm et al., 2017*). The cleaved 3' end, containing 2',3' cyclic phosphate, is trimmed by the 3'−5' RNA exosome to generate mature 5.8S RNA. Finally, the dynein-related AAA ATPase subunit, Mdn1, appears to mechanically remove biogenesis factors from the pre-60S particle to generate 60S ribosomal subunits (*Ulbrich et al., 2009*). Our findings suggest that, in addition to these essential functions, the rixosome is recruited to heterochromatin where its RNA processing activities are utilized to degrade heterochromatic RNAs in order to promote the spreading and epigenetic inheritance of H3K9me (*Figure 7A,B*).

The rixosome mutations described in this study abolish the DNA sequence-independent epigenetic inheritance of heterochromatin and, moreover, are completely defective in the spreading of H3K9me into a transgene inserted at a heterochromatic domain. These phenotypes are accompanied by accumulation of transgene RNA in rixosome mutants. Notably, the spreading defect only manifests itself at heterochromatin regions which lack a redundant RNA clearance pathway, namely RNAi. For example, at pericentromeric DNA repeats where the RNAi machinery is recruited to nascent noncoding RNAs and processes them into siRNAs, the rixosome contributes to noncoding RNA degradation, but is not required for H3K9me spreading, silencing, or inheritance – unless the RNAi pathway is rendered inactive by mutation. By contrast, at the boundaries of the mating type locus, where no redundant RNA degradation pathway operates, rixosome-mediated RNA clearance is essential for heterochromatin spreading and silencing. The rixosome thus defines a second RNA degradation mechanism that localizes to heterochromatin, through direct recruitment by Swi6, and plays an indispensable role in heterochromatin spreading and maintenance in transcription units that are not associated with RNAi (*Figure 7B*). Both the spreading and epigenetic inheritance of heterochromatin require the read-write capability of the Clr4 methyltransferase (*Zhang et al., 2008*; *Jih et al., 2017*; *Ragunathan et al., 2015*; *Audergon et al., 2015*). During read-write, Clr4 binds to nucleosomes containing pre-existing H3K9me and catalyzes the methylation of histone H3 in adjacent nucleosomes (*Zhang et al., 2008*; *Al-Sady et al., 2013*). Fission yeast heterochromatic domains are associated with low levels of transcription and the synthesized RNA has been suggested to be retained by Swi6 (*Keller et al., 2012*; *Bühler et al., 2007*; *Kloc et al., 2008*; *Chen et al., 2008*; *Jih et al., 2017*). The rixosome may cleave nascent pol II-associated RNAs or mature chromatin associated RNAs in the path of nucleosome-bound Clr4. In this model, in the absence of the rixosome, the RNA pol II complex or chromatin-associated RNAs would create a barrier that blocks Clr4 read-write (*Figure 7B*), which is required for both spreading of H3K9me from a nucleation point and its copying onto newly deposited nucleosomes following DNA replication.

In summary, the profound requirement for the rixosome in the spreading and epigenetic inheritance of H3K9me mirrors the requirement for the Clr4 read-write capability and supports the idea

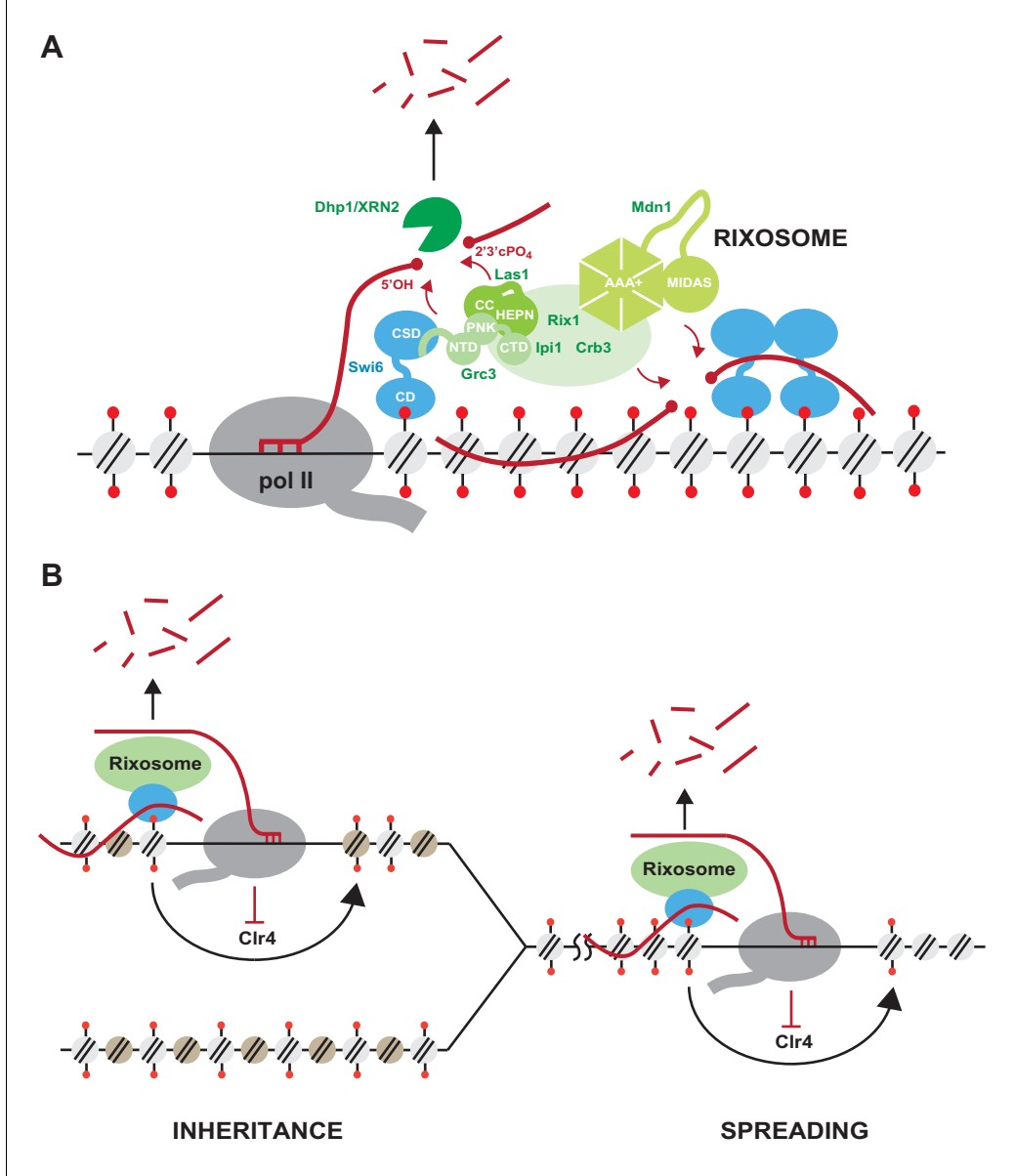

**Figure 7.** Model for rixosome-mediated degradation of nascent transcripts and its role in spreading and inheritance of heterochromatin. (**A**) The rixosome localizes to heterochromatin via a direct interaction between the N-terminus of Grc3 with Swi6. The endonuclease Las1 and the polynucleotide kinase (PNK) Grc3, subunits of the rixosome, cleave nascent (left, pol II-associated) and/or mature chromatin-associated (right) transcripts and phosphorylate the free 5' hydroxyl end at the site of cleavage, respectively. These processing steps prepare the RNA for subsequent degradation by the Dhp1/XRN2 exonuclease. CC, coiled coil domain; HEPN, higher eukaryotes and prokaryotes nucleotide-binding domain; AAA+, ATPase domain; pol II, RNA polymerase II. (**B**) Heterochromatic RNA clearance enables the spreading and inheritance of heterochromatin by Clr4-mediated read-write. Nascent transcripts and their associated transcriptional complexes, or mature RNAs retained at chromatin by heterochromatic proteins, present obstacles in the path of the read-write pathway. By degrading these RNAs, the rixosome facilitates read-write-dependent spreading and inheritance of heterochromatin.

that the rixosome and read-write act coordinately during spreading and inheritance of H3K9me. Our findings suggest that heterochromatin-associated rixosome acts as a key trigger that promotes RNA degradation by the Dhp1/XRN2 exoribonuclease. Finally, we note that subunits of the mammalian rixosome have been reported to interact with chromatin components and mutations in multiple sub-units of the complex are associated with human disease (*Fanis et al., 2012*; *Hussey et al., 2014*;

*Sareddy and Vadlamudi, 2016*). It will be important to understand the extent to which chromatin-dependent functions of the rixosome may contribute to the regulation of gene expression and development in mammals and other eukaryotes.

# Materials and methods

Key resources table

| Reagent type (species) or resource | Designation | Source or reference | Identifiers | Additional information |
|---|---|---|---|---|
| Strain, strain background (*Schizosaccharomyces pombe*) | various | This paper | | see *Supplementary file 1* |
| Antibody | Anti-H3K9me2 antibody (mouse monoclonal) | Abcam | Ab1220 | ChIP. 2 µg |
| Antibody | 8WG16 (mouse monoclonal) | Biolegend | MMS-126R-500 | ChIP, 4 µg |
| Antibody | Anti-H3K9me3 (monoclonal, recombinant) | Diagenode | C15500003 | ChIP, 1 µg |
| Antibody | Rabbit IgG (rabbit, polyclonal) | Sigma | 15006 | ChIP, 1.5 µg |
| Antibody | Peroxidase Anti-Peroxidase Soluble Complex antibody (rabbit) | Sigma | P1291 | Western, 1:2000 |
| Antibody | anti-Swi6 antibody (rabbit, polyclonal) | PMID:25730778 | | Western, 1:5000 |
| Antibody | IRDye 700DX-conjugated secondary antibody (goat, anti-rabbit) | VWR | RL611-130-122 | Western, 1:10000 |
| Sequence-based reagent | Oligonucleotides | This paper | | *Supplementary file 2* |
| Commercial assay or kit | ZR Fungal/ Bacterial DNA MiniPrep | Zymoresearch | 11–321 | |
| Commercial assay or kit | QIAquick PCR Purification Kit | Qiagen | 28106 | |
| Commercial assay or kit | Silver Stain SNAP Kit | Thermo Fisher Scientific | 24612 | |
| Commercial assay or kit | RNeasy Mini Kit | Qiagen | 74106 | |
| Commercial assay or kit | mirVana miRNA isolation kit | Ambion | AM1561 | |
| Commercial assay or kit | Qubit dsDNA high sensitivity kit | Invitrogen | Q32854 | |
| Chemical compound, drug | Ethyl methanesulfonate | Sigma Aldrich | M0880-5G | |
| Chemical compound, drug | Anhydrotetracycline (hydrochloride) | Thermo Fisher Scientific | 10009542100 MG | |

*Continued on next page*

*Continued*

| Reagent type (species) or resource | Designation | Source or reference | Identifiers | Additional information |
|---|---|---|---|---|
| Chemical compound, drug | 5-fluoroorotic acid | Goldbio | F-230–10 | |
| Chemical compound, drug | EGS (Ethylene glycol bis [succinimidylsuccinate]) | Pierce (Fisher) | 21565 | |
| Chemical compound, drug | Formaldehyde, 37% | Thermo Fisher Scientific | F79-500 | |
| Chemical compound, drug | PMSF | Thermo Fisher Scientific | 36978 | |
| Chemical compound, drug | cOmplete, EDTA-free Protease Inhibitor Cocktail Tablets | Sigma-Aldrich | 05056489001 | |
| Chemical compound, drug | Dynabeads Protein A | Invitrogen | 100-02D | |
| Chemical compound, drug | Dynabeads M-270 Epoxy | Invitrogen | 14301 | |
| Chemical compound, drug | Dynabeads M-280 Streptavidin 10 ml | Fisher | 11206D | |
| Chemical compound, drug | Dynabeads Pan Mouse IgG | Invitrogen | 110–41 | |
| Chemical compound, drug | trypsin | Promega | V5111 | |
| Chemical compound, drug | TMT-10plex reagent | Thermo Fisher Scientific | 90406 | |
| Chemical compound, drug | Superscript III reverse transcriptase | Invitrogen | 18080–044 | |
| Chemical compound, drug | TURBO DNase (2 U/µL) | Thermo Fisher | AM2239 | |
| Chemical compound, drug | 4–20% Mini-PROTEANÒTGX Precast Protein Gels | Biorad | 4561093 | |
| Chemical compound, drug | 0.5 mm glass beads 10 lbs bulk container | BioSpec | 11079105 | |
| Chemical compound, drug | Proteinase K | Sigma-Aldrich | 03115828001 | |
| Chemical compound, drug | Phenol:Chloroform: Isoamyl Alcohol 25:24:1 Saturated | Sigma-Aldrich | P2069-100ML | |
| Chemical compound, drug | Acid-Phenol: Chloroform, pH 4.5 | Thermo Fisher Scientific | AM9720 | |

*Continued on next page*

*Continued*

| Reagent type (species) or resource | Designation | Source or reference | Identifiers | Additional information |
|---|---|---|---|---|
| Software, algorithm | Mudi | PMID:24766403 | | T. Iida |
| Software, algorithm | Bowtie | https://doi.org/ 10.1186/gb- 2009-10-3-r25 | | |
| Software, algorithm | IGV | https://www. broadinstitute. org/igv/ | | |
| Software, algorithm | Geneious | http://www. geneious.com, PMID:22543367 | | |
| Software, algorithm | DeepTools | PMID:27079975 | | |

## Yeast strains and growth medium

The genotypes of all strains used in this study are presented in *Supplementary file 1*. Point mutations in the rixosome were reconstituted by seamless integration to avoid confounding effects from the presence of a selection cassette around the target genes. The genes of interest carrying the desired point mutations were cloned into a pFA6a plasmid backbone upstream of a *ura4+-hphMX6* cassette, amplified together with the selectable marker and transformed into yeast cells to replace the endogenous gene. The *ura4+-hphMX6* cassette was then knocked out by a second transformation to restore the native downstream region. All TAP (tandem affinity purification) tagged strains were expressed under the control of their endogenous promoters and terminators. Cells were cultured on rich YES (yeast extract with supplements) medium, unless otherwise indicated.

## Genetic screen for epigenetic inheritance mutants

Yeast cells carrying the *10xtetO-ade6+* reporter (*Ragunathan et al., 2015*) were grown in YES medium with shaking at 32°C to exponential phase (OD600 ~1), treated with 3% ethyl methanesulfonate (EMS) (*Ausubel and Winston, 2008*) and plated on low adenine YE medium at limiting dilution to form single colonies. The single colonies were replica plated on low adenine medium in the absence (YE) and presence of 5 µg/ml tetracycline (YE+TET). The replicas were incubated for 2 days at 32°C and scored for silencing phenotypes. Colonies which displayed silencing on YE medium (red) but not on YE+TET medium (white) were selected for further analysis. After three rounds of single colony purification, the mutants were backcrossed to the original reporter strain and the progeny was subjected to pooled linkage analysis, as described previously (*Birkeland et al., 2010*). Briefly, crosses were subjected to random spore analysis and 20–24 spores with the mutant phenotype were picked from YE plates. The mutant spores were cultured in individual wells in 24 well plates to saturation, pooled and collected. Genomic DNA was purified with the ZymoResearch Fungal/Bacterial DNA MiniPrep kit and sonicated using QSonica water bath sonicator (15 min with 10 s ON/OFF pulses at 60% amplitude) to obtain 200–500 bp fragment size distribution. Genomic DNA libraries were prepared according to a previously reported protocol (*Wilkening et al., 2013*) from the original reporter strain and the pools of mutant spores from each cross. Sequencing was performed on an Illumina HiSeq 2500 platform at a depth of at least 3 million reads per mutant pool. Read alignment, variant calling and annotation were performed using the Mudi platform (*Iida et al., 2014*).

## Growth and silencing assays

Yeast cells were grown in YES medium with shaking at 32°C to stationary phase (OD600 = 12 and above). The cultures were adjusted to a density of OD600 = 4 and serially diluted tenfold. Three microliters of each dilution were spotted on the appropriate medium. For assessing *ura4+* reporter gene silencing assays, serial dilutions were plated on EMMC-Ura (Edinburgh minimal medium lacking uracil) or YES medium supplemented with 5-fluoroorotic acid (0.1% and 0.2% 5FOA). For *ade6+*

reporter gene silencing assays, serial dilutions were plated on YE and YE+TET media. Phenotypes were scored after 3 days at 32°C.

## Chromatin immunoprecipitation (ChIP)

For TET treatment, yeast cells were grown for 24 hr in YES medium supplemented with TET to exponential phase (OD600 = 1–2). For all other ChIP experiments, yeast cultures were grown in rich (YES) medium to OD600 densities of 1–2. For Crb3-TAP ChIP, 50 OD600 units of cells per ChIP were fixed with 1.5% EGS for 45 min and 1% formaldehyde for 15 min, quenched with glycine and collected. For all other ChIPs, 50 OD600 units of cells were crosslinked with 1% formaldehyde for 15 min. Cell pellets were resuspended in lysis buffer (50 mM Hepes/KOH pH 7.5, 140 mM NaCl, 1 mM EDTA, 1% Triton-X-100, 0.1% Na Deoxycholate, 0.1% SDS, 1 mM PMSF, supplemented with protease inhibitors) and cells were lysed by the glass bead method, using MagNA Lyser (Roche) (3 pulses of 90 s at 4500 rpm, 1 pulse of 45 s at 5000 rpm). The lysates were sonicated by 3 pulses of 20 s (amplitude 40%) in a Branson microtip sonifier with intermittent cooling on ice, to a fragment distribution of 200–1000 bp. Cell debris was removed by centrifugation (16000xg) at 4*C for 15 min and the cleared lysates were incubated for 2 hr at 4*C with magnetic beads coupled to antibodies against the targets of interest. For H3K9me2 and RNA Pol II ChIPs, 30 µl Protein A Dynabeads (Invitrogen) were coupled with 2 µg anti-H3K9me2 antibody (Ab1220, Abcam) or 4 µg anti-Pol II (8WG16, Biolegend MMS-126R-500), respectively. For H3K9me3 ChIPs, 30 µl Streptavidin Dynabeads M280 (Invitrogen) were coupled to 1 µg of anti-H3K9me3 antibody (Diagenode C15500003) and blocked with 5 µM biotin. For Crb3-TAP ChIPs, 4.5 µg Dynabeads M-270 Epoxy beads (Invitrogen) were coupled to 1.5 µg rabbit IgG (Sigma, #15006) and incubated with the cleared lysate. The bead-protein complexes were washed three times with lysis buffer, once with chilled TE buffer (10 mM Tris/HCl, 1 mM EDTA) and eluted using 120 µl 50 mM Tris/HCl pH 8.0, 1 mM EDTA, 1% SDS with heating (65°C) for 20 min. Crosslinks were reversed at 65°C overnight and the eluates were treated with RNase A and Proteinase K (Roche), prior to an extraction with phenol:chloroform:isoamyl alcohol. qPCR was performed on Applied Biosystems qPCR instrument with the primer sequences listed in *Supplementary file 2*. qPCR quantification and statistical analysis were performed for three biological replicates.

## Library preparation and next generation sequencing

For ChIP-seq, reverse crosslinked DNA treated with RNase A and proteinase K was purified using the Qiagen PCR purification kit. The DNA was then sonicated in a QSonica water bath sonicator (5 min with 15 s ON/OFF pulses, 20% amplitude) to a fragment size of ~200 bp, concentrated and quantified using Qubit dsDNA high sensitivity kit. 1–10 ng of DNA were used to prepare libraries as described previously (*Wong et al., 2013*). Libraries were pooled and sequenced on Illumina HiSeq and NextSeq platforms. The raw reads were demultiplexed using Geneious and aligned to the reference genome using bowtie with random assignment of reads to repeats. The mapped reads were normalized to counts per million using Deeptools and visualized in the IGV genome browser. Sequencing data were deposited in the Gene Expression Omnibus (GEO) under the accession number GSE140920.

## Immunoprecipitation of rixosome complexes

Purifications of Crb3-TAP tagged rixosome complexes were performed by a chromatin purification method described previously (*Iglesias et al., 2020*) with minor modifications. Cells carrying Crb3-TAP in the wild-type and mutant (*grc3-V70M or crb3-D198N*) backgrounds, as well as corresponding untagged cells, were grown in rich YES medium to exponential phase (OD600 = 1–1.5) and 600 OD600 units were collected per immunoprecipitation. The pellets were resuspended in lysis buffer (20 mM Hepes pH7.5-NaOH, 100 mM NaCl, 5 mM MgCl2, 1 mM EDTA pH 8.0, 10% Glycerol, 0.25% Triton-X-100, 0.5 mM DTT, 2 mM phenylmethylsulphonylfluoride [PMSF], protease inhibitor cocktail). The cells were lysed by the glass bead method in a MagNA Lyser (Roche) with a succession of short pulses (10 cycles, 22 s at 5000 rpm, 9 cycles, 15 s, 5500 rpm), followed by rapid cooling for 2 min in an ice water bath. Lysates were clarified at 16,000xg for 15 min and incubated with 300 µl of Pan Mouse IgG Dynabeads (Invitrogen) per IP. Four washes were performed with lysis buffer with full resuspension of the beads, followed by four short washes with wash buffer (20 mM Hepes pH7.5-

NaOH, 100 mM NaCl, 5 mM MgCl2). Proteins were eluted with 0.5M NH4OH and dried down in a speed vac. Five to ten percent of each elution fraction was analyzed by silver staining using the Silver Stain SNAP Kit (Pierce) according to the manufacturer's protocol. The remainder of each elution was analyzed by mass spectrometry.

## Mass spectrometry sample preparation and data analysis

All liquid reagents used for mass spectrometry sample preparation were HPLC grade. Proteins were digested with trypsin (Promega #V5111) in 200 mM HEPES buffer pH 8.5 with 2% acetonitrile (v/v) overnight at 37°C. For direct TMT peptide labeling of digests, TMT-10plex reagents from Thermo Fisher Scientific (#90406) were used. After quenching with hydroxylamine at 0.3% (v/v), reactions were acidified with formic acid and peptides purified by reversed phase $C_{18}$ chromatography via stage tips. Data were collected on an Orbitrap Lumos instrument with a multi-notch $MS^3$ method. Prior to injection, peptides were separated by HPLC with an EASY-nLC 1000 liquid chromatography system (Thermo Scientific) with a 100 μm inner diameter column and a 2.6 μm Accucore $C_{18}$ matrix (Thermo Fisher Scientific) with a 4 hr acidic acetonitrile gradient. $MS^1$ scans were recorded in the Orbitrap (resolution 120,000, mass range 400–1400 Th) and $MS^2$ scans in the iontrap after collision-induced dissociation (CID, CE = 35) and a maximum injection time of 150 ms. For TMT quantification, an SPS-$MS^3$ method was used with HDC fragments scanned in the Orbitrap at a resolution of 50,000 at 200 Th with a maximum injection time of 200 ms. Peptides were searched with an in-house software based on SEQUEST (v.28, rev. 12) against a forward and reverse database of the *S. pombe* proteome database (Uniprot) with human contaminants added. Searches had a mass tolerance of 50 ppm for precursors and a $MS^2$ fragment ion tolerance of 0.9 Da. For searches, 2 missed cleavages were allowed and methionine oxidation was searched for dynamically (+15.9949 Da). After linear discriminant analysis (LDA), peptide false discovery rate was 1%, as was the FDR for final collapsed proteins. $MS^1$ data were post-search, calibrated, and the search performed again. For TMT quantification, peptides with an isolation specificity of >70% and a summed TMT signal-to-noise (s/n)>200 were filtered. Proteins with at least two peptide quantification events were considered for analysis. For all 1139 quantified proteins, an enrichment score was calculated as the summed TMT signal-to-noise (TMT sum s/n) ratio of the tagged wild-type sample over the untagged TMT sum s/n of the wild-type control. The top 10% most enriched proteins (118) contained all known components of the Rix1 core complex (Crb3/Ipi3, Rix1/Ipi2, Ipi1, Las1, Grc3, Mdn1), many of the pre-60S ribosome particle components (ribosomal and ribosome biogenesis proteins), Swi6 (required for the recruitment of the Rix1 complex to heterochromatin) and nuclear export factors known to be involved in the shuttling of the pre-60S complex to the cytosol. The wild-type/mutant fold change score for each protein was calculated as the ratio of the summed TMT s/n in the wild-type over the mutant IPs, after subtraction of the background from the untagged control and correction for the abundance of the bait protein. The log transformed fold-change scores were plotted against the log transformed enrichment scores to identify proteins which were over- or under-represented in the mutant sample relative to the wild-type. Z scores were calculated from the distribution of fold-change scores of the top 10% most enriched proteins and the Z-scores of outliers were displayed on the plot.

## Western blot

To prepare whole cell protein extracts, yeast cells were grown in YES medium at 32°C with shaking to exponential phase (OD600 = 1) and 25 ml of culture (25 OD600 units) were collected. Cells were resuspended in lysis buffer (50 mM Tris pH 7.5, 150 mM NaCl, 5 mM EDTA, 10% glycerol, 1 mM PMSF, protease inhibitor cocktail) and lysed by the glass bead method in a MagNA Lyser (Roche) with a succession of short pulses (5 pulses, 45 s at 4500 rpm), followed by rapid cooling in an ice water bath. Aliquots of 0.25 and 0.5 OD600 units were resolved by SDS-PAGE electrophoresis, immunoblotted with anti-TAP antibody (PAP, Sigma P1291) and anti-Swi6 antibody (rabbit polyclonal, *Holoch and Moazed, 2015b*), followed by IRDye 700DX-conjugated secondary antibody (LI-COR Biosciences), and imaged on an Odyssey imaging system (LI-COR Biosciences).

## Total RNA extraction and reverse transcription

Yeast cultures were grown to exponential phase (OD600 = 1) and 50 OD600 units of cells were collected. Cells were lysed by the hot phenol method, RNA was purified on-column using RNeasy Mini Kit (Qiagen), treated with Turbo DNase (ThermoFischer) and cleaned up by a second RNeasy Mini kit purification. 500 ng of pure RNA was reverse transcribed (Superscript III kit, Invitrogen) using the primers indicated in *Supplementary file 2* and quantified by qPCR on Applied Biosystems qPCR instrument. $act1^+$ was used as an internal control. Statistical analysis was performed on three biological replicates.

## rRNA processing

Yeast cultures were grown to exponential phase (OD600 = 1) and 50 OD600 units of cells were collected. Cells were lysed by the hot phenol method, RNA was precipitated with ethanol and resuspended in water. RNA was separated by denaturing gel electrophoresis in a MOPS-formaldehyde buffer system and visualized by staining with EtBr.

## sRNA northern blot

Yeast cultures were grown in YES medium to exponential phase (OD600 = 1), a cell count was determined by a BrightLine hemacytometer and $5 \times 10^7$ cells were collected. sRNAs were purified with the mirVana miRNA isolation kit (Ambion) and 25 µg of sRNA were used for Northern blotting as described previously (*Bühler et al., 2006*). Sequences of oligo probes are listed in *Supplementary file 2*.

## Acknowledgements

We thank Tessi Iida for help and advice in analysis of whole genome sequencing data for identification of mutants, members of the Moazed lab for helpful discussions, Swapnil Parhad, Antonis Tatarakis, Xiaoyi Wang, Andy Yuan, Chen Zhou, and Haining Zhou for comments on the manuscript, and Nahid Iglesias for Swi6 purification protocols. This work was supported by an HHMI summer student internship award (AD), a fellowship from the Boehringer Ingelheim Fonds (GS), and NIH RO1 GM072805 (DM). DM is a Howard Hughes Medical Institute Investigator.

## Additional information

### Funding

| Funder | Grant reference number | Author |
| --- | --- | --- |
| National Institutes of Health | RO1 GM072805 | Danesh Moazed |
| Howard Hughes Medical Institute | | Danesh Moazed |
| Boehringer Ingelheim Fonds | Fellowship | Gergana Shipkovenska |
| Howard Hughes Medical Institute | Summer student internship award | Alexander Durango |

The funders had no role in study design, data collection and interpretation, or the decision to submit the work for publication.

### Author contributions

Gergana Shipkovenska, Conceptualization, Investigation, Methodology, Writing - original draft, Writing - review and editing; Alexander Durango, Marian Kalocsay, Investigation, Writing - review and editing; Steven P Gygi, Supervision; Danesh Moazed, Conceptualization, Supervision, Funding acquisition, Writing - original draft, Writing - review and editing

### Author ORCIDs

Gergana Shipkovenska (iD) https://orcid.org/0000-0001-6288-4401
Danesh Moazed (iD) https://orcid.org/0000-0003-0321-6221

Decision letter and Author response
Decision letter https://doi.org/10.7554/eLife.54341.sa1
Author response https://doi.org/10.7554/eLife.54341.sa2

## Additional files

### Supplementary files

• Supplementary file 1. List of *S. pombe* strains used in this study. This Table outlines the list of fission yeast *S. pombe* strains that were used in this study and notes the specific figure in the paper in which each strain was used.

• Supplementary file 2. List of oligonucleotides used in this study. List of DNA oligonucleotides used in RT-qPCR, ChIP-qPCR and northern blots.

• Supplementary file 3. The results of quantitative mass spectrometry experiments for the purification of Grc3-V70M mutant rixosome complexes. List of proteins identified in purifications of wildtype and Grc3-V70M mutant rixosome complexes using isobaric tandem mass tags (TMT) mass spectrometry.

• Supplementary file 4. The results of quantitative mass spectrometry experiments for the purification of Crb3-D198N mutant rixosome complexes. List of proteins identified in purifications of wildtype and Crb3-D198N mutant rixosome complexes using isobaric tandem mass tags (TMT) mass spectrometry.

• Transparent reporting form

### Data availability

Sequencing data have been deposited in GEO under accession code GSE140920.

The following dataset was generated:

| Author(s) | Year | Dataset title | Dataset URL | Database and Identifier |
| --- | --- | --- | --- | --- |
| Danesh M, Gergana S | 2019 | An RNA Endonuclease-Kinase Complex Required for Spreading and Epigenetic Inheritance of Heterochromatin | https://www.ncbi.nlm.nih.gov/geo/query/acc.cgi?acc=GSE140920 | NCBI Gene Expression Omnibus, GSE140920 |

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
