## [Decision Letter]

**Acceptance summary:**

This work identifies the rixosome, a RNA degradation complex, that regulates heterochromatin spreading and inheritance in fission yeast. The authors discover the intersection of this RNA degradation by the rixosome and canonical RNAi pathway to ensure heterochromatin spreading. Overall this work identifies important mechanisms for the interplay between RNA and heterochromatin.

**Decision letter after peer review:**

Thank you for submitting your article "A Conserved RNA Degradation Complex Required for Spreading and Epigenetic Inheritance of Heterochromatin" for consideration by *eLife*. Your article has been reviewed by three peer reviewers, one of whom is a member of our Board of Reviewing Editors, and the evaluation has been overseen by James Manley as the Senior Editor. The following individual involved in review of your submission has agreed to reveal their identity: John L Rinn (Reviewer #2).

The reviewers have discussed the reviews with one another and the Reviewing Editor has drafted this decision to help you prepare a revised submission.

Summary:

Moazed and colleagues report the identification of rixosome, a RNA degradation complex previously implicated in rRNA biogenesis , that is important for heterochromatin spreading and inheritance in fission yeast. The authors conduct a forward genetic screen with an inducible heterochromatin system, and show that several hits comprise a RNA degradation complex they term the rixosome. Rixosome interacts with Clr4 H3K9 methylase and generates RNA fragments that are substrates for XRN2/Dhp1 exonuclease. Additional functional and genetic interaction studies show that the rixosome functions redundantly with RNAi pathway to ensure heterochromatin spreading. Altogether, this study expands on previous findings (Kitano et al., 2011; Chalamcharla et al., 2015; Tucker et al., 2016) and brings them into a new and exciting perspective. Overall this work identifies important mechanisms for the interplay between RNA and heterochromatin, and should be published after the following points are clarified.

Essential revisions:

1) Interplay between rixosome and RNAi needs to be clarified. On one hand, authors claim that these pathways compete and bring as evidence increase of sRNA in *grc3-V70M* mutant (Figure 4H lane 2). On the other hand, the authors argue that rixosome and RNAi work together in a redundant manner for heterochromatin spreading and inheritance. E.g. in Figure 2E and F *ago1Δ* seem to increase the effect of rixosome mutation suggesting a cooperative action in the maintenance of H3Kme at endogenous heterochromatic domain. Similarly Figure 5 and 6 argue for the redundancy model. The confusion arises from how rixosome has specificity for which RNA. If rixosome only cleaves euchromatin RNA, then RNAi and rixosome could work together. If rixosome cleaves heterochromatic RNA that is necessary for RNAi-dependent heterochromatin, then rixosome would counteract RNAi. Please address experimentally or comment on whether rixosome has specificity toward different classes of RNAs.

2) Another approach the authors took for testing a post-transcriptional role of RIXC in silencing are ChIP experiments with RNA Pol II. These revealed only mild (i.e. 2-fold: *caf1grc3-V70M* vs. *caf1* at *10xtetO-ura-gfp*; Figure 4F) or no increase of RNA Pol II (grc3-V70 vs. wild type at *L(BglII):ade6*; Figure 5F), which let the authors conclude that transcription is not affected. However, also RNA levels were only moderately increased in those mutants, and even clr4 mutants did not show changes for RNA Pol II for one of the reporters (Figure 5F). Thus, rather than ruling out transcriptional changes, these RNA Pol II ChIP experiments appear to be not sufficiently sensitive. A better approach for measuring transcription and RNA turnover would be NET-seq and pulse-labeling with 4-thiouridine (4SU) (Mata and Wise, 2017), respectively. Otherwise, these statements should be toned down.

3) By northern blots, the authors showed increased sRNA in *grc3-V70M* and *dhp1-1* single and double mutants, which seems to be consistent with their model. However, no increase was seen for *swi6* and the corresponding double mutant *swi6grc3-V70M*. These results seem inconsistent and raise doubts whether this assay is quantitative and reliable. Experiments should be repeated and sRNA levels should be quantitatively assessed, and appropriate loading controls that can also be quantified (other than SybrGold stain) should be included.

---

## [Author Response]

Essential revisions:1) Interplay between rixosome and RNAi needs to be clarified. On one hand, authors claim that these pathways compete and bring as evidence increase of sRNA in grc3-V70M mutant (Figure 4H lane 2). On the other hand, the authors argue that rixosome and RNAi work together in a redundant manner for heterochromatin spreading and inheritance. E.g. in Figure 2E and F ago1Δ seem to increase the effect of rixosome mutation suggesting a cooperative action in the maintenance of H3Kme at endogenous heterochromatic domain. Similarly Figure 5 and 6 argue for the redundancy model. The confusion arises from how rixosome has specificity for which RNA. If rixosome only cleaves euchromatin RNA, then RNAi and rixosome could work together. If rixosome cleaves heterochromatic RNA that is necessary for RNAi-dependent heterochromatin, then rixosome would counteract RNAi. Please address experimentally or comment on whether rixosome has specificity toward different classes of RNAs.

We apologize for any confusion regarding the relationship between the rixosome and the RNAi pathway. Thank you for explicitly describing the source of the confusion. Our findings clearly indicate that the 2 pathways can act on the same RNA substrates at the pericentromeric repeats (as the reviewer implies, the pathways act in parallel or compete for the same RNA substrates). Thus, when rixosome function at heterochromatin is eliminated, more RNA substrate is available for processing by the RNAi machinery, resulting in higher siRNA levels in rixosome mutant cells (new Figure 5G, lane 2). Conversely, in the absence of RNAi, the residual H3K9me at the pericentromeric repeats becomes entirely rixosome-dependent, indicating that the rixosome, and by implication its RNA degradation activity, is required for H3K9me maintenance in the absence of RNAi (Figure 2F, G).

The data presented in our manuscript clearly demonstrate that the rixosome is *not* necessary for RNAi-dependent heterochromatin, as indicated by the observations that (1) RNAi-dependent H3K9me and silencing at pericentromeric DNA repeats is not affected by rixosome mutations [Figure 2 and Figure 2—figure supplement 2] and (2) mutations in the rixosome result in increased pericentromeric siRNA levels [Figure 2G]. We have strived to modify the text and figure legends to make these points clearer.

2) Another approach the authors took for testing a post-transcriptional role of RIXC in silencing are ChIP experiments with RNA Pol II. These revealed only mild (i.e. 2-fold: caf1 grc3-V70M vs. caf1 at 10xtetO-ura-gfp; Figure 4F) or no increase of RNA Pol II (grc3-V70 vs. wild type at L(BglII):ade6; Figure 5F), which let the authors conclude that transcription is not affected. However, also RNA levels were only moderately increased in those mutants, and even clr4 mutants did not show changes for RNA Pol II for one of the reporters (Figure 5F). Thus, rather than ruling out transcriptional changes, these RNA Pol II ChIP experiments appear to be not sufficiently sensitive. A better approach for measuring transcription and RNA turnover would be NET-seq and pulse-labeling with 4-thiouridine (4SU) (Mata and Wise, 2017), respectively. Otherwise, these statements should be toned down.

We thank the reviewer for this comment but believe that our presentation of the RNA Pol II ChIP and the qRT data may have led to a misunderstanding. As the reviewer states, we see only modest changes in RNA Pol II occupancy in *grc3-V70M* versus wild type for 2 different reporter genes at 2 different heterochromatic regions [*10xtetO::ura4-gfp* and *L(BglII):ade6^+^* loci](new Figures 5D, E and Figure 6F). Regarding the possibility that “RNA Pol II ChIP experiments appear to be not sufficiently sensitive”, we believe that the data we had already presented rules out this possibility for the first locus. In Figure 5E, we included a crucial control in which heterochromatin is absent and the reporter gene is maximally derepressed (tetR-clr4-I, *clr4Δ*). Comparison of tetR-clr4-I, *clr4Δ* mutant with wild type (clr4+) control in Figures 5E shows that our RNA Pol II ChIP signal at this locus displays a large dynamic range of ~120 fold. Despite little or no change in RNA Pol II occupancy in the *caf1Δgrc3-V70M* double mutant cells (Figure 5E), we observe a 5-fold increase in RNA levels (Figure 5C). It is important to note that these experiments were performed under establishment conditions in which we show that the rixosome is not required for silencing (Figure 1D). The data show that even under conditions in which the reporter gene is strongly silenced, we could detect some increase in RNA levels in the absence of rixosome-mediated RNA degradation, which supports the model that the RNA processing/degradation activities of the rixosome are required for heterochromatin maintenance. In the revised manuscript, we provide further support for this idea by demonstrating that the catalytic polynucleotide kinase activity of the Grc3 subunit of the complex – which prepares RNA substrates for degradation by Dhp1/XRN2 – is required for heterochromatin maintenance (new Figure 4).

Regarding modest increases in RNA Pol II occupancy and RNA levels in rixosome and clr4 mutants at the mating type locus (new Figure 6E and F), we failed to note that some transcriptional repression at the mating locus occurs in a heterochromatin-independent manner. It has previously been shown that a heterochromatin-independent (Clr4- and H3K9me-independent) mechanism represses the expression of the *mat2P* gene (Hansen et al., PLOSgenetics 2011, cited in the revised manuscript). Our results are therefore consistent with previous findings and indicate that loss of rixosome localization to the mating type locus has little or no effect on the H3K9me-independent repression of mat2P.

We were unable to more fully address the possible lack of sensitivity in detection of changes in RNA Pol II occupancy noted by the reviewer due to shutdown of laboratories to combat COVID19 spreading. We are eager to perform the nascent transcript experiments mentioned by the reviewers in the near future. We have removed the section in the Discussion where we proposed that the at some target genes [such as *L(BglII)::ade6^+^*] the rixosome may function in silencing by promoting RNA degradation without an effect on RNA Pol II occupancy. We note that whether or not the rixosome affects RNA Pol II occupancy at target loci, our conclusion that its RNA degradation activities are critical for heterochromatin spreading and epigenetic inheritance remain valid.

3) By northern blots, the authors showed increased sRNA in grc3-V70M and dhp1-1 single and double mutants, which seems to be consistent with their model. However, no increase was seen for swi6 and the corresponding double mutant swi6 grc3-V70M. These results seem inconsistent and raise doubts whether this assay is quantitative and reliable. Experiments should be repeated and sRNA levels should be quantitatively assessed, and appropriate loading controls that can also be quantified (other than SybrGold stain) should be included.

As requested by the reviewers, we quantified the siRNA data for the appropriate mutant cells and used *snoR69 and a spike-in 20mer RNA* as loading controls for the northern blots (new Figure 5G and H and Figure 5—figure supplement 1C-E).

We apologize for not describing these results more clearly. As noted by the reviewer, we observed increased sRNA levels in *grc3-V70M* and dhp1-1 single and double mutants, which is consistent with a role for rixosome- and Dhp1-dependent RNA degradation at pericentromeric DNA repeats. We note that the absence of an effect of the *grc3-V70M* in *swi6Δ* cells is exactly what we would expect because (1) in *swi6Δ* cells, the rixosome is no longer recruited to pericentromeric DNA repeats [Iglesias et al., 2020, Supplementary Figure 1] and (2) the *grc3-V70M* mutation disrupts the interaction of the rixosome with Swi6 and its localization to heterochromatin (this study, Figure 3). We would therefore not expect the *grc3-V70M* mutant to have any effect on dg siRNA levels as each mutation (*swi6Δ* and *grc3-V70M*) abolishes the localization of the rixosome to heterochromatin, i.e., they act in the same pathway.

Finally, we would like to note that the lack of an increase in siRNA levels in cells lacking Swi6 is an expected result. We and others have previously shown that Swi6 stabilizes the RNAi machinery on chromatin and is required, to a variable degree depending on the locus, for accumulation of siRNAs. Therefore, although in swi6D cells, the rixosome is no longer recruited to heterochromatin, in these cells, siRNAs do not accumulate to the same level as swi6+grc3D or swi6+dhp1D cells. We have now included this explanation in the text (page 16, lines 368-372).